# Learning What Reinforcement Learning Can't: Interleaved Online Fine-Tuning for hardest Questions

**Lu Ma**[1], **Hao Liang**[1,2], **Meiyi Qiang**[1], **Lexiang Tang**[1], **Xiaochen Ma**[1], **Zhen Hao Wong**[1]
**Junbo Niu**[1], **Chengyu Shen**[1], **Runming He**[1], **Yanhao Li**[1], **Bin Cui**[1,4,*], **Wentao Zhang**[1,2,3*]
[1]Peking University, [2]Zhongguancun Academy
[3]Beijing Key Laboratory of Data Intelligence and Security (Peking University)
[4]Beijing Key Laboratory of Software and Hardware Cooperative Artificial Intelligence Systems

## Abstract

Recent advances in large language model (LLM) reasoning have shown that reasoning ability can emerge through reinforcement learning (RL). However, despite these successes, RL in its current form remains insufficient to induce capabilities that exceed the limitations of the base model, as it is primarily optimized based on existing knowledge of the model. To address this limitation, we employ supervised fine-tuning (SFT) to learn what RL cannot, which enables the incorporation of new knowledge and reasoning patterns by leveraging high-quality demonstration data. We analyze the training dynamics of RL and SFT for LLM reasoning and find that RL excels at improving performance on questions within the model's original capabilities, while SFT is more effective at enabling progress on questions beyond the current scope of the model. Motivated by the complementary strengths of RL and SFT, we introduce **ReLIFT** (**Re**inforcement **L**earning **I**nterleaved with Online **F**ine-**T**uning), a novel training strategy. ReLIFT employs RL for general training, but interleaves it with targeted SFT on challenging questions for which high-quality solutions are collected online. By alternating between RL and SFT, ReLIFT addresses model weaknesses as they emerge. Empirically, ReLIFT outperforms previous RLVR methods by an average of +6.7 points across a suite of six benchmarks (five math reasoning and one out-of-distribution). More importantly, ReLIFT surpasses baselines such as individual RL, individual SFT, and various hybrid approaches while reducing the required training time. These results provide compelling evidence that ReLIFT is a powerful and resource-efficient paradigm for developing capable reasoning models. The code is available at here.

## 1 Introduction

Recent advancements in large language model reasoning, as demonstrated by OpenAI o-series Jaech et al. (2024), DeepSeek-R1 Guo et al. (2025), Gemini 2.5 Comanici et al. (2025), and Kimi k-series Team et al. (2025b), highlight the potential to enhance reasoning capabilities by increasing test-time computational resources. These models typically generate long Chains-of-Thought (CoT Wei et al. (2022)) responses, exhibiting sophisticated behaviors such as reflection Kumar et al. (2024) and planning Ke et al. (2025). The primary factor driving this progress is large-scale Reinforcement Learning with Verifiable Rewards (RLVR), which does not depend on traditional techniques like Monte Carlo Tree Search Xie et al. (2024); Li et al. (2025) or Process Reward Models Setlur et al. (2024); Zhang et al. (2025). In RLVR, rewards are assigned based on whether the model's output matches a ground-truth solution in mathematics or passes unit tests in code, thereby enabling scalability without the need for demonstration data. Implemented with policy optimization algorithms like Proximal Policy Optimization (PPO) Schulman et al. (2017) and Group Relative Policy Optimization (GRPO) Shao et al. (2024), the simplicity and directness of large-scale RLVR have been pivotal in incentivizing the reasoning capabilities seen in today's most advanced LLMs.

---

*Corresponding authors: Bin Cui and Wentao Zhang.

Despite its empirical successes, RLVR is insufficient to foster capabilities that transcend a base model's inherent limitations Yue et al. (2025); Cheng et al. (2025); Zhao et al. (2025). Research suggests that RLVR primarily reinforces existing behaviors rather than instilling novel reasoning abilities. For instance, Yue et al. (2025) argue that RLVR fails to equip LLMs with new reasoning skills, noting that RLVR-trained models underperform base models on pass@k at higher k values, indicating limited expansion of reasoning capabilities. Similarly, Zhao et al. (2025) show that RL post-training amplifies certain pre-training behaviors while suppressing others. Furthermore, Cheng et al. (2025) find that RL stifles exploration, causing models to converge on narrow behavioral patterns and plateau on complex reasoning tasks. This limitation is rooted in the on-policy nature of RL, which learns from the model's own generated responses via iterative rollouts and feedback. Consequently, RL primarily enhances performance by biasing the model toward reasoning paths it already knows are likely to yield rewards, optimizing existing knowledge rather than facilitating the acquisition of new information.

In contrast, supervised fine-tuning (SFT) offers a complementary approach to enhancing the reasoning capabilities of LLMs. By leveraging high-quality demonstration data, SFT facilitates the incorporation of new knowledge and reasoning patterns Mecklenburg et al. (2024); Yue et al. (2025); Guo et al. (2025), and has been shown to achieve superior reasoning performance compared to RL, particularly in small-scale LLMs Guo et al. (2025). Nevertheless, the effectiveness of SFT relies highly on the availability of substantial amounts of high-quality demonstration data. Furthermore, SFT-trained models often exhibit limited generalization to out-of-distribution (OOD) scenarios, while RL-based approaches tend to demonstrate greater robustness Chu et al. (2025); Chen et al. (2025). These contrasting characteristics of RL and SFT highlight a promising research direction:

> *How can RL and SFT be effectively combined to improve LLM reasoning and OOD generalization, reduce dependence on expensive demonstration data, and achieve capabilities beyond current cognitive constraints?*

To achieve this goal, we analyze the training dynamics of RL and SFT by examining how the accuracy of questions changes throughout the training process. We evaluate the model's accuracy on questions at various checkpoints, and classify them into four levels of difficulty. Our findings indicate that RL is more effective for questions of lower difficulty, whereas SFT proves more beneficial for the most challenging ones. Furthermore, for simpler questions, SFT can degrade the model's existing performance and the response length increases. In contrast, for challenging questions, the improvement provided by RL is less pronounced compared to SFT. Overall, RL efficiently trains the model to generate correct answers for questions it can already solve, while SFT is crucial for enabling the model to address questions that surpass its current capabilities.

Based on these analyses, we propose **Re**inforcement **L**earning **I**nterleaved with Online **F**ine-**T**uning (**ReLIFT**). As illustrated in Figure 2, during RL training, we collect highly challenging examples according to the accuracy observed during the rollout. When such a challenging question is identified, we obtain a high-quality CoT solution and subsequently filter out any incorrect answers. After constructing these examples, we add them to an SFT buffer. Once the number of challenging questions in the SFT buffer is sufficient, we perform SFT on these questions for one step.

We conduct comprehensive experiments on five challenging math reasoning benchmarks and one OOD benchmark. Our proposed ReLIFT method achieves a new state-of-the-art accuracy of 52.6% with Qwen2.5-Math-7B, which significantly outperforms previous RLVR baselines. Notably, ReLIFT demonstrates clear superiority over methods including pure SFT, pure RL, and combined RL and SFT approaches, such as RL with SFT loss, SFT followed by RL, and LUFFY Yan et al. (2025). ReLIFT requires minimal detailed demonstration data and GPU training hours. Additionally, ReLIFT generates significantly more concise solutions, which improves both performance and efficiency. We further validate the generality of ReLIFT by extending it to smaller and weaker base models, consistently observing superior results.

The main contributions of this work are as follows:

- We present a systematic analysis of the training dynamics of RL and SFT in reasoning, empirically demonstrating their complementary roles: RL refines existing skills on solvable problems, while SFT is essential for acquiring new knowledge for difficult ones.

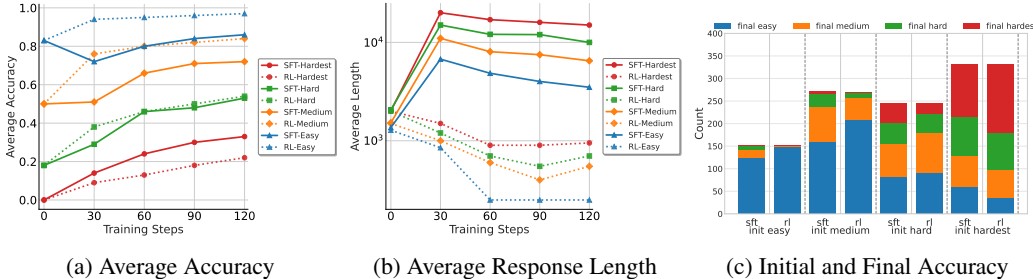

(a) Average Accuracy     (b) Average Response Length     (c) Initial and Final Accuracy

Figure 1: Accuracy and response length changes for *Easy*, *Medium*, *Hard*, and *Hardest* questions during RL and SFT training. (a) Average accuracy change for each difficulty category. (b) Average response length change for each difficulty category. (c) Number of questions transitioning between different initial and final accuracy categories in RL and SFT, respectively. The x-axis represents the initial difficulty category, and the y-axis represents the final difficulty category.

- We propose ReLIFT, a novel framework that interleaves RL with online fine-tuning. ReLIFT dynamically identifies challenging examples during RL for targeted fine-tuning to address emerging model weaknesses.

- Extensive experiments show ReLIFT achieves state-of-the-art performance on mathematical reasoning and OOD benchmarks, outperforming pure SFT, pure RL, and existing hybrid methods with significantly less demonstration data and computational overhead.

## 2 REINFORCEMENT LEARNING INTERLEAVED WITH ONLINE FINE-TUNING

### 2.1 RL VS. SFT: TRAINING DYNAMICS ACROSS QUESTION DIFFICULTY LEVELS

We begin our study by examining how the model's accuracy evolves on questions of varying difficulty during both SFT and RL training. Specifically, we independently train the Qwen2.5-Math-7B model using both SFT and RL on a subset of the Open-R1-math-220K dataset Face (2025). Both SFT and RL are conducted for 120 steps, with model checkpoints saved every 30 steps. Detailed experimental settings are provided in Appendix B. At each checkpoint, we evaluate model performance on a validation set comprising 1,000 questions. To facilitate a more detailed analysis, we categorize the validation questions into four distinct difficulty levels: *Easy*, *Medium*, *Hard*, and *Hardest*. To determine the difficulty level of each question $q$, we prompt the Qwen2.5-Math-7B model eight times and calculate the average accuracy for that question, denoted as $acc(q)$. Following the approach in works like Wen et al. (2025); Bae et al. (2025); Yu et al. (2025), and to ensure a sufficient number of questions in each category, we select these specific thresholds to enable a robust analysis of the training dynamics of both RL and SFT. The criteria for assigning difficulty levels are as follows:

- *Easy*: 6 to 8 correct answers out of 8 ($acc(q) \geq 0.75$).
- *Medium*: 3 to 5 correct answers out of 8 ($0.375 \leq acc(q) \leq 0.625$).
- *Hard*: 1 or 2 correct answers out of 8 ($0.125 \leq acc(q) \leq 0.25$).
- *Hardest*: 0 correct answers out of 8 ($acc(q) = 0$), representing items that are beyond the model's current capabilities.

We analyze the average accuracy of LLM trained with RL and SFT on four difficulty levels of questions: *Easy*, *Medium*, *Hard*, and *Hardest*. As shown in Figure 1a, RL outperforms SFT on *Easy* and *Medium* questions. Conversely, SFT achieves better results than RL on the *Hardest* questions. Notably, after SFT, the LLM fails to solve certain questions that the initial model previously answered correctly. In contrast, RL consistently improves accuracy across all difficulty levels and does not exhibit this issue. We also examine changes in the average response length during RL and SFT training, as shown in Figure 1b. SFT produces longer responses across all question difficulties, aiming to mimic the target long response. In contrast, RL begins with shorter responses, and for *Medium* and *Hard* questions, the response length increases over time, indicating more exploration. For the

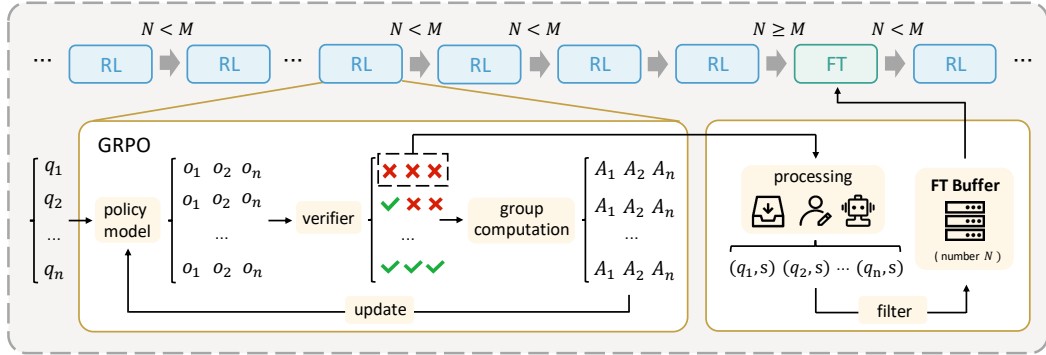

Figure 2: **Overview of the ReLIFT Training Framework.** The model is mainly trained with RL. When it encounters particularly hard questions, high-quality solutions are collected or generated, then stored in a buffer. Once enough hard examples are gathered, a fine-tuning (FT) step is performed using these examples. This process adaptively alternates between RL and FT to help the model learn from its mistakes and improve reasoning ability. In addition, $N$ denotes the number of *hardest* $(q, s)$ pairs in the buffer, while $M$ represents a predefined threshold, typically set to the batch size for the FT.

*Hardest* questions, the response length remains stable, which suggests the model stops exploring further. Ideally, responses to difficult questions should be longer, and responses to simple questions should be shorter. However, neither SFT nor RL fully achieves this objective.

We further analyze the evolution of accuracy during training by tracking both the initial and final accuracy for each question, as illustrated in Figure 1c. Our findings reveal substantial differences in how RL and SFT influence the model's performance across questions of varying difficulty. For *Easy* and *Medium* questions, SFT results in decreased accuracy for some items, whereas RL generally preserves or enhances the LLM's original ability to solve these questions. However, this pattern reverses for *Hardest* questions: SFT enables the model to learn and solve a greater number of these challenging items, as evidenced by the increased transition of *Hardest* questions into other categories. In contrast, RL does not yield comparable improvements for *Hardest* questions, highlighting a limitation in its ability to support learning beyond its current capabilities.

These findings demonstrate that RL and SFT have complementary strengths. RL excels at refining and preserving the model's existing knowledge, making it suitable for questions within its competence. However, SFT is superior for acquiring new knowledge to tackle problems that lie beyond the model's initial capabilities. This suggests that accuracy does not improve uniformly and that a hybrid training strategy could be optimal. By selectively interleaving SFT steps for the *Hardest* questions, it may be possible to enhance overall model performance and develop more scalable and effective RL for LLMs.

## 2.2 Reinforcement Learning Interleaved with Online Fine-Tuning

**Overview.** Motivated by the complementary strengths of RL and SFT in addressing questions of varying difficulty, we propose a novel training approach termed **Re**inforcement **L**earning **I**nterleaved with Online **F**ine-**T**uning (**ReLIFT**). ReLIFT dynamically integrates RL and fine-tuning (FT) based on the characteristics of the training data and the current capabilities of the model, as illustrated in Figure 2. During RL training, we identify *hardest* examples based on rollout accuracy. For each of these *hardest* questions, we obtain high-quality CoT solutions either by collecting them in advance before training or by consulting a stronger model or human annotators. These curated examples are added to a fine-tuning buffer. When the buffer accumulates enough questions to form a training batch, we perform a fine-tuning step using these examples. The underlying intuition is that questions difficult for the model to learn through RL alone may benefit from targeted fine-tuning. Below, we provide a detailed description of our method.

**Reinforment Learning while Collecting Hard Questions.** Owing to the success of Deepseek-R1, GRPO has become the de facto approach for RLVR training. GRPO leverages the reward scores of

$N$ sampled solutions from a query to estimate the advantage, thereby eliminating the need for an additional value model. Formally, we denote the policy model before and after the update as $\pi_{\theta_{\text{old}}}$ and $\pi_\theta$. Given a question $q$, a set of solutions $o_i$ generated by $\pi_{\theta_{\text{old}}}$, and the reward function $R(\cdot)$, the GRPO objective is defined as follows:

$$\mathcal{L}_{\text{GRPO}}(\theta) = \frac{1}{\sum_{i=1}^{N} |o_i|} \sum_{i=1}^{N} \sum_{t=1}^{|o_i|} \min \left[ r_{i,t}(\theta) A_i, \text{clip} \left( r_{i,t}(\theta); 1 - \epsilon, 1 + \epsilon \right) A_i \right], \quad (1)$$

where $r_{i,t}(\theta) = \dfrac{\pi_\theta \left( o_{i,t} \mid q, o_{i,<t} \right)}{\pi_{\theta_{\text{old}}} \left( o_{i,t} \mid q, o_{i,<t} \right)}, A_i = \dfrac{R \left( o_i \right) - \text{mean} \left( \{ R \left( o_i \right) \mid o_i \sim \pi_{\theta_{\text{old}}}(o), i = 1, 2, \ldots, N \} \right)}{\text{std} \left( \{ R \left( o_i \right) \mid o_i \sim \pi_{\theta_{\text{old}}}(o), i = 1, 2, \ldots, N \} \right)}.$

During the rollout phase of GRPO, for each question $q$, we generate a set of solutions $o_i$, which are evaluated by the reward function to obtain the accuracy $acc(q)$. We identify particularly challenging questions as those for which $acc(q) = 0$. For these particularly hard questions, we solicit high-quality CoT solutions $s$ from a stronger reasoning model (e.g., DeepSeek-R1) or human experts, filter out incorrect CoTs to obtain reliable QA pairs $(q, s)$, and add them to a buffer $\text{Buffer}_{\text{hardest}}$. The data collected from $\text{Buffer}_{\text{hardest}}$ is then added to a primary fine-tuning buffer, $\text{Buffer}_{\text{FT}}$, which serves as the data for subsequent fine-tuning step. This process is performed online during training. Collecting and training can be done together, and can be formalized as:

$$\text{Buffer}_{\text{hardest}} = \left\{ (q, s) \mid \text{acc}(q) = 0, \ s = M(q), \ \text{extract}(s) = a \right\}, \quad (2)$$

$$\text{Buffer}_{\text{FT}} \leftarrow \text{Buffer}_{\text{FT}} \cup \text{Buffer}_{\text{hardest}},$$

where $M(q)$ denotes the CoT response obtained either in advance or online from an external model or human annotator, $\text{extract}(s)$ indicates the final answer extracted from $s$, and $a$ is the groundtruth answer for $q$. We only keep $(q, s)$ pairs where $\text{extract}(s) = a$.

**Interleaved Online Fine-Tuning for Hardest Questions**. When the size of the fine-tuning buffer exceeds a predefined threshold $M$ (i.e., $|\text{Buffer}_{\text{FT}}| \geq M$), we sample a batch of $M$ $(q, s)$ pairs from the buffer and perform a fine-tuning step using these hard examples. The fine-tuning step minimizes the standard cross-entropy loss:

$$\mathcal{L}_{\text{CE}}(\theta) = -\frac{1}{|s|} \sum_{i=1}^{|s|} \log \pi_\theta \left( s_i \mid q, s_{<i} \right), \quad \text{where } (q, s) \in \text{Buffer}_{\text{FT}}. \quad (3)$$

To prevent fine-tuning updates from overly constraining the model's exploratory behavior, we incorporate an entropy regularization term into the loss function. The full objective is as follows:

$$\begin{aligned}
\mathcal{L}_{\text{FT}}(\theta) &= \mathcal{L}_{\text{CE}}(\theta) + \alpha \mathcal{L}_{\text{Entropy}}(\theta) \\
&= -\frac{1}{|s|} \sum_{i=1}^{|s|} \log \pi_\theta \left( s_i \mid q, s_{<i} \right) - \alpha \frac{1}{|s|} \sum_{i=1}^{|s|} H(\pi_\theta(s_i \mid q, s_{<i})),
\end{aligned} \quad (4)$$

where $(q, s) \in \text{Buffer}_{\text{FT}}$. The term $H(\pi_\theta(s_i \mid q, s_{<i}))$ represents the entropy of the token generation distribution $\pi_\theta(s_i \mid q, s_{<i})$, and $\alpha$ is a hyperparameter controlling the weight of the entropy term.

The frequency of this interleaved fine-tuning is adaptive. Early in training, when the model's performance is low, SFT is applied more frequently to accelerate the learning of effective reasoning patterns. As training progresses and the model improves, RL is prioritized to further incentivize the model's existing abilities. Furthermore, this online approach obviates the need for a large, pre-collected dataset of CoT demonstrations $s$. Instead, CoTs can be generated dynamically only for the most challenging questions encountered during training.

## 3 Experimental Setup

**Dataset Construction.** Our training set is a subset of OpenR1-Math-220k Face (2025), with prompts sourced from NuminaMath 1.5 Li et al. (2024) and detailed demonstration data generated by Deepseek-R1 Guo et al. (2025). We use the default 94k prompt subset and filter out generations that are longer than 8192 tokens or are verified incorrect by Math-verify Face (2024). This process yields 46k prompts and high-quality demonstrations, which we term *OpenR1-Math-46k-8192*.

**Evaluation.** For our evaluation, we primarily consider five widely adopted mathematical reasoning benchmarks: AIME 2024, AIME 2025, AMC Li et al. (2024), OlympiadBench He et al. (2024), and MATH500 Hendrycks et al. (2021). Due to the relatively small test sets for AIME 2024, AIME 2025, and AMC, we report avg@32 for these datasets. For OlympiadBench and MATH500, we use avg@8 as the evaluation metric. Since our reinforcement learning training is mainly centered on mathematical reasoning, we also assess generalization performance on MMLU-Pro Wang et al. (2024). To prevent data contamination, we shuffle the multiple-choice options. A testing temperature of 0.6 was used for all evaluations.

**Baselines.** We benchmark ReLIFT against the following baselines using **Qwen2.5-Math-7B** Yang et al. (2024) as the base model. For SFT methods, we fine-tuning the base model on the *OpenR1-Math-46k-8192* dataset. For RL methods, we include GRPO Shao et al. (2024) trained on the same 46k dataset; **SimpleRL-Zero** Zeng et al. (2025), which applies GRPO to approximately 24k mathematical samples from GSM8K Cobbe et al. (2021) and MATH dataset Hendrycks et al. (2021); **OpenReasoner-Zero** Hu et al. (2025), a PPO-based approach trained on 129k multi-source samples; **PRIME-Zero** Cui et al. (2025), which conducts policy rollouts on 150k NuminaMath queries with implicit process rewards; and **Oat-Zero** Liu et al. (2025), which removes the standard deviation in GRPO advantage calculation, and is trained on the MATH dataset. For methods combining SFT and RL, we evaluate three approaches: **RL w/ SFT loss**, which incorporates a SFT loss term into the GRPO objective trained on the 46k dataset; **LUFFY** Yan et al. (2025), a mixed-policy GRPO approach that also utilizes the 46k dataset; and **SFT then RL**, a two-stage method that applies RL training after an initial SFT phase. Specifically, we evaluate two versions of SFT then RL: **SFT then RL (v1)** involves taking our trained SFT model and training it with RL for 300 steps on a held-out split of the OpenR1-Math-220k dataset, which contains about 49,000 prompts, similar to previous work Cui et al. (2025); Yan et al. (2025). For **SFT then RL (v2)**, to fairly compare with **ReLIFT**, which uses 8,640 demonstration samples, we first perform SFT on a subset of 8,640 samples from *OpenR1-Math-46k-8192*. Then we conduct GRPO training on the full *OpenR1-Math-46k-8192* dataset. For a more detailed setup for training these baselines, please refer to Appendix B.

## 4 Experimental Results

### 4.1 Main Results

**SOTA performance with Qwen2.5-Math-7B.** Table 1 presents our main results, comparing ReLIFT against four previous state-of-the-art RLVR methods and replication of SFT and RL, all based on the Qwen2.5-Math-7B model. Across five challenging competition-level reasoning benchmarks and one out-of-distribution benchmark, ReLIFT establishes a new state-of-the-art with an overall accuracy of **52.6%**, surpassing all baselines. Notably, ReLIFT consistently achieves either the best or second-best performance on every individual benchmark, indicating ReLIFT achieves both higher reasoning performance and better generalization. More OOD results can be found in the Appendix G.

Table 2: Comparison of resource requirements of methods.

| Model | GPU Hours | Demonstrations |
|---|---|---|
| SFT | $8 \times 8$ | 46K |
| RL | $40 \times 8$ | 0 |
| RL w/ SFT loss | $113.5 \times 8$ | 46K |
| SFT then RL v1 | $57 \times 8$ | 46K |
| SFT then RL v2 | $63.5 \times 8$ | 8K |
| LUFFY | $73 \times 8$ | 46K |
| ReLIFT | $52 \times 8$ | 8K |

**Comparing against methods combining SFT and RL.** When compared with SFT and RL methods trained on the same dataset, ReLIFT outperforms all methods that combine RL and SFT, including RL w/SFT loss, SFT then RL, and LUFFY, as shown in Table 1. ReLIFT also produces more concise responses. Additionally, an analysis of the required demonstration data and GPU hours in Table 2

Table 1: Overall performance on five math benchmarks and one OOD benchmark based on Qwen2.5-Math-7B. Bold, single underline, and double underline represent the best, second-best, and third-best. *ACC* represents the average accuracy. *LEN* represents the average number of tokens.

| Model | AIME-24 | | AIME-25 | | AMC | | MATH-500 | | Olympiad | | MMLU-Pro | | Overall | |
|---|---|---|---|---|---|---|---|---|---|---|---|---|---|---|
| | ACC | LEN | ACC | LEN | ACC | LEN | ACC | LEN | ACC | LEN | ACC | LEN | ACC | LEN |
| **Qwen2.5-Math-7B** | | | | | | | | | | | | | | |
| **Qwen-Math** | 16.4 | 2125 | 6.9 | 2039 | 45.5 | 1493 | 65.1 | 1171 | 29.7 | 1594 | 24.9 | 830 | 31.4 | 1542 |
| **Qwen-Math-Instruct** | 9.3 | 4065 | 8.2 | 3590 | 40.5 | 2905 | 78.2 | 1774 | 36.2 | 2888 | 34.0 | 4274 | 34.4 | 3249 |
| **Previous RLVR methods** | | | | | | | | | | | | | | |
| **SimpleRL-Zero** | 27.0 | - | 6.8 | - | 54.9 | - | 76.0 | - | 34.7 | - | 34.5 | - | 39.0 | - |
| **OpenReasoner-Zero** | 16.5 | - | 15.0 | - | 52.1 | - | 82.4 | - | 47.1 | - | **58.7** | - | 45.3 | - |
| **PRIME-Zero** | 17.0 | - | 12.8 | - | 54.0 | - | 81.4 | - | 40.3 | - | 32.7 | - | 39.7 | - |
| **Oat-Zero** | **33.4** | - | 11.9 | - | 61.2 | - | 78.0 | - | 43.4 | - | 41.7 | - | 45.9 | - |
| **Replication of SFT and RL** | | | | | | | | | | | | | | |
| **SFT** | 26.9 | 7344 | **25.5** | 7059 | 59.8 | 5675 | 84.8 | 3503 | 52.6 | 5662 | 44.4 | 3956 | 49.0 | 5533 |
| **RL** | 21.1 | 3287 | 17.5 | 2904 | 62.1 | 2127 | 85.7 | 1277 | 48.6 | 2178 | 46.3 | 1276 | 46.9 | 2175 |
| **Methods combining SFT and RL** | | | | | | | | | | | | | | |
| **RL w/ SFT loss** | 26.9 | 7344 | 23.1 | 7059 | 59.8 | 5675 | 84.1 | 3503 | 53.6 | 5662 | 44.4 | 3956 | 48.6 | 5508 |
| **SFT then RL (v1)** | 29.4 | 5581 | 21.0 | 5328 | 63.1 | 3908 | 87.3 | 2224 | 55.7 | 3772 | 47.6 | 2258 | 50.7 | 3845 |
| **SFT then RL (v2)** | 26.8 | 6453 | 23.0 | 6231 | 63.8 | 4586 | **88.1** | 2682 | 54.4 | 4615 | 48.9 | 2638 | 50.8 | 4534 |
| **LUFFY** | 27.3 | 5741 | 23.0 | 5267 | 63.5 | 3477 | 85.6 | 2361 | 53.8 | 3758 | 52.6 | 2241 | 50.9 | 3808 |
| **ReLIFT** | 28.3 | 5062 | 22.9 | 4571 | **65.1** | 3540 | 87.9 | 2241 | **57.3** | 3352 | 53.9 | 2248 | **52.6** | 3502 |

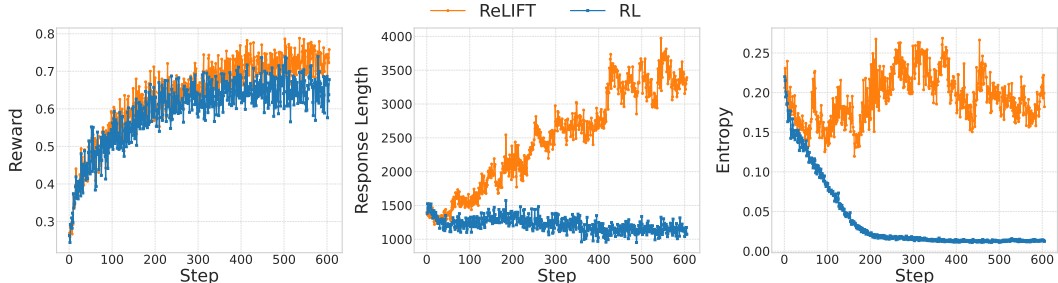

Figure 3: Training Dynamic of rewards, response lengths, and the training entropy during RL and ReLIFT training.

reveals that ReLIFT requires less training time and demonstration samples while achieving the best performance. These results underscore the effectiveness and efficiency of ReLIFT.

## 4.2 TRAINING DYNAMICS

Figure 3 presents the training dynamics of rewards, response lengths, and training entropy during RL and ReLIFT training. The reward achieved by ReLIFT consistently surpasses that of RL, suggesting that ReLIFT is more effective in optimizing the target objective. Regarding response lengths, RL tends to generate shorter responses as training progresses, which may reflect a diminishing capacity to address challenging questions. In contrast, ReLIFT demonstrates a gradual increase in response length, implying a higher potential to solve difficult problems. The entropy of the RL model steadily declines throughout training, indicating reduced exploration. In contrast, the entropy of ReLIFT remains relatively high and exhibits persistent fluctuations, consistently surpassing that of RL. This sustained exploration allows ReLIFT to continue discovering novel solutions, thereby facilitating ongoing performance improvements even in the later stages of training.

Table 3: Overall performance on five math benchmarks and one OOD benchmark based on Qwen2.5-Math-1.5B and Qwen2.5-7B. Bold and underline indicate the best and second-best results, respectively.

| Model | AIME-24 | | AIME-25 | | AMC | | MATH-500 | | Olympiad | | MMLU-Pro | | Overall | |
|---|---|---|---|---|---|---|---|---|---|---|---|---|---|---|
| | *ACC* | *LEN* | *ACC* | *LEN* | *ACC* | *LEN* | *ACC* | *LEN* | *ACC* | *LEN* | *ACC* | *LEN* | *ACC* | *LEN* |
| **Qwen2.5-Math-1.5B** | | | | | | | | | | | | | | |
| **Qwen-Math-1.5B** | 2.8 | 1971 | 1.3 | 2058 | 19.5 | 1533 | 45.4 | 1108 | 18.5 | 1619 | 5.2 | 1982 | 15.5 | 1712 |
| **Qwen-Math-1.5B-Instruct** | 10.3 | 4042 | 7.6 | 3764 | 40.0 | 2809 | 77.8 | 1628 | 35.7 | 2834 | 33.7 | 4300 | 34.2 | 3230 |
| **SFT** | 12.7 | 7925 | **13.0** | 7719 | 40.9 | 6953 | 71.8 | 5986 | 33.8 | 7231 | 23.5 | 5405 | 32.6 | 6870 |
| **RL** | 9.8 | 2425 | 8.5 | 2195 | 44.1 | 1569 | 74.2 | 958 | 37.2 | 1695 | 31.4 | 1075 | 34.2 | 1653 |
| **ReLIFT** | **14.3** | 3691 | 10.0 | 3416 | **47.2** | 2207 | **76.4** | 1308 | **39.6** | 2274 | **31.7** | 1724 | **36.5** | 2437 |
| **Qwen2.5-7B** | | | | | | | | | | | | | | |
| **Qwen-7B** | 6.2 | 1632 | 2.4 | 1370 | 31.1 | 1076 | 63.8 | 617 | 26.5 | 1068 | 32.8 | 945 | 27.1 | 1118 |
| **Qwen-7B-Instruct** | 11.5 | 1939 | 6.1 | 1500 | 41.0 | 1629 | 73.0 | 1753 | 37.3 | 1550 | 53.1 | 3325 | 37.0 | 1949 |
| **SFT** | 15.7 | 7786 | **18.6** | 7533 | 49.8 | 6424 | 80.8 | 4252 | 44.3 | 6234 | 54.6 | 4869 | 44.0 | 6183 |
| **RL** | 15.5 | 1784 | 13.0 | 1487 | 50.4 | 1422 | 80.0 | 917 | 42.2 | 1438 | **57.6** | 954 | 43.1 | 1334 |
| **ReLIFT** | **19.1** | 5522 | 14.7 | 4831 | **51.9** | 3792 | **81.6** | 2211 | **46.1** | 3649 | 56.4 | 2173 | **45.0** | 3696 |
| **LLaMA-3.1-8B** | | | | | | | | | | | | | | |
| **LLaMa-8B-Instruct** | **5.6** | 1396 | 0.7 | 1401 | **20.7** | 1012 | **44.0** | 622 | **13.8** | 1063 | 38.1 | 357 | **20.5** | 975 |
| **SFT** | 0.8 | 2033 | **1.5** | 2032 | 11.5 | 2007 | 28.6 | 1903 | 8.6 | 1997 | 28.2 | 1558 | 13.2 | 1922 |
| **RL** | 1.8 | 940 | 0.0 | 909 | 10.8 | 797 | 28.2 | 616 | 7.3 | 790 | 39.4 | 549 | 14.6 | 767 |
| **ReLIFT** | 1.3 | 1236 | 0.2 | 1272 | 11.9 | 1048 | 35.2 | 741 | 11.0 | 1050 | **44.2** | 650 | 17.3 | 1000 |

### 4.3 ABLATION STUDY AND ANALYSIS

We conduct an ablation study to understand what makes ReLIFT effective. Our findings reveal that the scheduling of fine-tuning and the selection of training data are crucial. Specifically, we compare the following three ablation settings: ReLIFT(all), which alternates one fine-tuning step after each RL step on the same batch, with the number of demonstrations matching SFT; ReLIFT(uniform), which applies fine-tuning at fixed, uniform intervals (every 8 RL steps), with the number of demonstrations matching ReLIFT; ReLIFT(random) which performs interleaved fine-tuning. The buffer is populated based on the number of *Hardest* questions, but instead of using the hardest questions, it is randomly filled with non-hardest questions.

As shown in Figure 4, ReLIFT (all) quickly collapses, probably due to conflicting optimization objectives between RL and FT. Similarly, using a uniform or random schedule for FT, as in ReLIFT(uniform) and ReLIFT(random), leads to lower accuracy and longer responses. In contrast, ReLIFT strategically applies fine-tuning more frequently in early training stages and focuses on the most challenging questions. This approach allows the model to efficiently gain new knowledge without sacrificing performance or conciseness. More detailed results are in Appendix D.

We also study the effect of the entropy coefficient, $\alpha$, on the fine-tuning step, using Qwen2.5-Math-7B. As seen in Figure 5, an $\alpha$ of 0.0001 prove optimal,

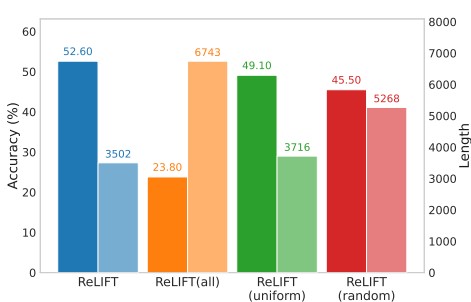

Figure 4: Ablation study on ReLIFT. The left bar and the right bar represents average accuracy and length, respectively.

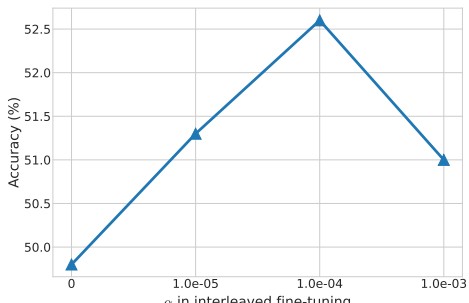

Figure 5: Accuracy versus the choice of $\alpha$.

achieving the highest accuracy of 52.6. Varying $\alpha$ in either direction from this value significantly decreased performance. These results highlight the importance of the entropy loss term for effectively blending SFT with RL. Therefore, we use $\alpha = 1 \times 10^{-4}$ consistently in our experiments.

## 4.4 EXTENSION TO MORE MODELS

To evaluate the generalizability of our method, we extend ReLIFT to three additional base models: the smaller Qwen2.5-Math-1.5B Yang et al. (2024), the weaker Qwen2.5-7B Team (2024), and Llama3.1-8b Dubey et al. (2024), which has a different architecture. As shown in Table 3, ReLIFT consistently achieves substantial improvements across all base models and six benchmarks, surpassing both SFT and RL. Our method consistently achieves a top-two ranking while maintaining an acceptable response length. These results underscore the broad applicability of ReLIFT in achieving superior performance and generalization across diverse model scales and architectures.

## 5 RELATED WORKS

### 5.1 RL-DRIVEN EMERGENCE OF REASONING ABILITIES

RL has recently emerged as a promising approach for enhancing the reasoning capabilities of LLMs. Early breakthroughs, such as OpenAI o-series Jaech et al. (2024); OpenAI (2025), DeepSeek-R1 Guo et al. (2025), and Kimi k-series Team et al. (2025b;a), have demonstrated that applying RLVR can lead to significant improvements in complex reasoning tasks. Notably, DeepSeek-R1 further shows that applying RLVR directly to a base model, without the need for intermediate SFT. Building on these foundations, Light-r1 Xie et al. (2025) incorporates curriculum learning to refine LLM's reasoning skills, while Logic-rl Xie et al. (2025) adopts logic-driven reward functions to improve general reasoning ability. Deepscaler Luo et al. (2025) trains LLMs with progressively longer contexts as their performance improves. Based on GRPO, DAPO Yu et al. (2025) uses clip-higher to prevent entropy collapse, dynamic sampling for improved efficiency, and token-level policy gradient loss with overlong reward shaping to stabilize training.

### 5.2 TRAINING PARADIGMS

Despite these significant advances, Yue et al. (2025) argue that RLVR does not enable LLMs to acquire genuinely new reasoning abilities beyond those present in their base models, as RLVR-trained models perform worse on pass@k than the base models at higher k values, indicating limited reasoning capability expansion. Similarly, Zhao et al. (2025) demonstrate that RL creates an "echo chamber" effect, amplifying specific behavioral patterns already present in the pre-training data while suppressing others, thereby reinforcing existing behaviors rather than creating new reasoning capabilities. Furthermore, Cheng et al. (2025) finds that RL stifles exploration, causing models to converge on narrow behaviors and leading to performance plateaus on complex reasoning tasks. Research suggests that current RLVR methods do not fully leverage the potential of RL to create genuinely novel reasoning abilities in LLMs. In contrast, directly fine-tuning on demonstration data introduces new knowledge to LLMs. However, SFT requires high-quality, detailed demonstration data and tends to encourage memorization of the training data, often resulting in poor generalization to out-of-distribution scenarios Chu et al. (2025); Chen et al. (2025); Team et al. (2025b).

Some approaches attempt to combine SFT and RL. Typically, these methods first use SFT to stabilize the LLM's ability and then apply RL to unlock more advanced reasoning skills in LLMs Guo et al. (2025); Wen et al. (2025). However, such combined methods remain limited: SFT depends significantly on the quality of demonstration data, and the subsequent RL phase is still confined to the knowledge present in the SFT-trained model, making it difficult for the model to acquire genuinely new information. LUFFY Yan et al. (2025) seeks to address these limitations by augmenting RL with off-policy reasoning traces, dynamically balancing imitation and exploration through a combination of off-policy demonstrations and on-policy rollouts during training. Nevertheless, this approach necessitates the preparation of off-policy data in advance and relies heavily on probability-sharpening techniques, which may restrict its applicability. In contrast, ReLIFT proposes a simpler and more direct method that integrates RL and SFT to leverage the strengths of both paradigms. Our approach encourages LLMs to explore when possible and integrate new information as needed, thereby pushing RL-driven reasoning beyond the boundaries of existing knowledge.

# 6 DISCUSSION AND CONCLUSION

## 6.1 DISCUSSION

**Data Source and Core Method:** In our experiments, we use DeepSeek-R1 as the source for high-quality data, a decision made for reproducibility and cost-effectiveness. This does not imply that ReLIFT is a distillation method. The core mechanism of ReLIFT—identifying "hardest" problems online and interleaving FT—is source-agnostic. As mentioned in the paper, the framework is equally applicable to other data sources, such as human expert annotators. Due to resource constraints, this study does not include experiments related to human annotation; this task is reserved for future research. Therefore, based solely on the experiments presented here, the approach is considered a distillation method. However, the scalability of our method, owing to its low requirement for demonstration data, maintains the feasibility of using human annotation.

**Feasibility of Online Collection:** While our experiments use offline-collected data due to resource constraints, a true online collection is technically feasible. The key is that the RL policy update step is time-intensive, which is sufficient to "mask" the latency of external data collection. The system can asynchronously dispatch identified hard problems to an API or human queue while the RL update is being computed. As long as the data generation rate matches the RL's problem identification rate, online collection would not slow overall training.

## 6.2 CONCLUSION

In summary, we systematically analyze the respective strengths of RL and SFT for LLM reasoning, finding that RL is more effective for learning easier questions while SFT is crucial for solving challenging questions. Building on these insights, we introduce ReLIFT, an approach that interleaves RL with online fine-tuning on the hardest questions. ReLIFT achieves state-of-the-art results with significantly less demonstration data, reduced training time, and more concise solutions. Future work will focus on scaling ReLIFT to larger models and developing more effective strategies for coordinating SFT and RL to further advance LLM reasoning and generalization.

## ACKNOWLEDGMENTS

This work was supported by the National Natural Science Foundation of China (Grant Nos. 92470121, 62402016, U23B2048, U22B2037), the National Key R&D Program of China (Grant No. 2024YFA1014003), the Fundamental and Interdisciplinary Disciplines Breakthrough Plan of the Ministry of Education of China (Grant No. JYB2025XDXM108), Zhongguancun Academy (Grant Nos. C20250204, C20250602), and the High-Performance Computing Platform of Peking University.

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

# A  THE USE OF LARGE LANGUAGE MODELS

We use LLMs to polish our writing. They act as a sophisticated assistant, improving the clarity, grammar, and style of our text.

# B  MAIN EXPERIMENT DETAILS

**RL Implementation.** For RL, we omit the KL divergence loss term following Luo et al. (2025); Yan et al. (2025). Consistent with Dr.GRPO Liu et al. (2025) and LUFFY Yan et al. (2025), we exclude both length normalization and standard error normalization from the GRPO loss 1 in all experiments. The rollout batch size is set to 128, the number of rollout is 8, the update batch size is 64, and the learning rate is $1 \times 10^{-6}$. A temperature of 1.0 is used during rollout generation for exploration. For Qwen2.5-Math-7B, We train 600 steps for all methods except SFT then RL, to be near with the GPU hour of ReLIFT. For other base models, we train 500 steps. All experiments are conducted on 8 x A800 GPUs.

**SFT Implementation.** For all our SFT models, we utilize the *OpenR1-Math-46k-8192* dataset. Following a hyperparameter search on a sampled small-scale dataset, we evaluate batch sizes of 64, 128, and 256, in conjunction with learning rates of $1 \times 10^{-5}$, and $5 \times 10^{-5}$. The optimal configuration identified includes a batch size of 64 and a learning rate of $5 \times 10^{-5}$. Employing the LLaMA-Factory framework Zheng et al. (2024), we train our models for 3 epochs.

**RL w/ SFT Loss Implementation.** We compute the on-policy loss on 7 on-policy samples and SFT loss on 1 off-policy sample per prompt. The remaining setup is consistent with our RL experiments.

**SFT then RL Implementation.** We conduct SFT using the same settings outlined in the SFT Implementation and train for 3 epochs accordingly. Subsequently, the RL phase follows the same configuration as described in RL Implementation, and we train the RL phase for 300 steps.

**ReLIFT Implementation.** During interleaved fine-tuning, we make use of demonstrations collected beforehand. The RL and fine-tuning configurations for ReLIFT are consistent with the settings previously described except learning rate. For both the RL and interleaved SFT phases, the learning rates are set to $1 \times 10^{-6}$.

**Evaluation.** All evaluations are conducted using VLLM Kwon et al. (2023), with the temperature set to 0.6 and the maximum number of tokens set to 8192. Results are verified using MATH-VERIFY Face (2024). For the baselines, we independently reproduce and verify the results of all methods combining SFT and RL, such as LUFFY Yan et al. (2025), whereas the results for previous RLVR methods are reported as in the LUFFY paper.

**Reward Function.** To evaluate the impact of our method, we adopt a simple reward function as below. All training experiments employ the same reward function.

$$r = \begin{cases} 1, & \text{if the answer is correct} \\ 0, & \text{otherwise} \end{cases}$$

**Qwen2.5-Math Models.** All evaluations are conducted using VLLM Kwon et al. (2023), with a temperature setting of 0.6 and a maximum token limit of 8192. Results are validated using MATH-VERIFY Face (2024). For baseline methods, we independently reproduce and verify results for all approaches combining SFT and RL, including LUFFY Yan et al. (2025). The results for other RL models are reported following the findings presented in the LUFFY paper.

**Llama-3.1-8B.** Because the full *OpenR1-Math-46k-8192* dataset proved too challenging for Llama-3.1-8B, we created a smaller, more manageable dataset. We selected 11,000 samples from OpenR1 where DeepSeek-R1 provided the correct solution and the solution length was under 2,048 tokens. The SFT was then performed using a learning rate of $2 \times 10^{-5}$, as suggested by Liu et al. (2025). The maximum response length is set to 2048 tokens for RL training and evaluation.

**Motivation Experiment Details.** We randomly sample a subset of 8,000 examples from the training dataset described in Section 3 as the training set, and select 1,000 examples as the validation set. RL and SFT are both conducted on this subset dataset for 120 steps, using the same parameters as in our main experiments.

## C  CHAT TEMPLATE

To encourage the base model to reason more effectively, following Yan et al. (2025), we adopt a complex chat template as shown in Figure 6. All our trained models, except LLaMA-3.1-8B, share the same system prompt for training and inference.

---

**Qwen Chat Template**

Your task is to follow a systematic, thorough reasoning process before providing the final solution. This involves analyzing, summarizing, exploring, reassessing, and refining your thought process through multiple iterations. Structure your response into two sections: **Thought** and **Solution**.

In the **Thought** section, present your reasoning using the format: "<think>\n {thoughts} </think>\n". Each thought should include detailed analysis, brainstorming, verification, and refinement of ideas.

After "</think>\n," in the **Solution** section, provide the final, logical, and accurate answer, clearly derived from the exploration in the Thought section.

If applicable, include the **Answer** in \boxed{} for closed-form results like multiple choices or mathematical solutions.

User: This is the problem: {Question}

Assistant: <think>

---

Figure 6: Chat template for Qwen-series.

---

**Llama Chat Template**

Question: {Question}.

Answer: Let's think step by step.

---

Figure 7: Chat template for Llama3.1-8B.

For LLaMA-3.1-8B, we do not use the above system prompt as we find the model cannot follow such an instruction. Thus, we use a simplified version as shown in Figure 7.

## D  DETAILED EXPERIMENT RESULTS

Table 4: Ablation study on ReLIFT. Bold indicate the best result.

| Model | AIME-24 | | AIME-25 | | AMC | | MATH-500 | | Olympiad | | MMLU-Pro | | Overall | |
|---|---|---|---|---|---|---|---|---|---|---|---|---|---|---|
| | ACC | LEN | ACC | LEN | ACC | LEN | ACC | LEN | ACC | LEN | ACC | LEN | ACC | LEN |
| **ReLIFT** | **28.3** | 5062 | **22.9** | 4571 | **65.1** | 3540 | 87.4 | 2241 | **57.8** | 3352 | **53.9** | 2248 | **52.6** | 3502 |
| **ReLIFT(all)** | 7.8 | 7647 | 9.0 | 7648 | 31.6 | 6653 | 54.2 | 5320 | 26.4 | 6600 | 13.7 | 6590 | 23.8 | 6743 |
| **ReLIFT(uniform)** | 23.2 | 6113 | 19.3 | 5318 | 62.3 | 4016 | 86.0 | 1971 | 52.4 | 2926 | 51.4 | 1952 | 49.1 | 3716 |
| **ReLIFT(random)** | 22.1 | 4616 | 13.9 | 4676 | 59.5 | 5271 | 84.2 | 5820 | 44.7 | 5187 | 48.4 | 6020 | 45.5 | 5268 |

The comprehensive results for ReLIFT(all), ReLIFT(uniform), and ReLIFT(random) are presented in Table 4. These findings indicate that merely alternating between RL and fine-tuning is insufficient; both the scheduling and the selection of fine-tuning training data are crucial components for success.

## E  REASONING BEHAIVERS

To analyze the distinct behaviors of the RL, SFT, and ReLIFT models, we first curate a pool of keywords corresponding to three key cognitive actions: summarizing, rethinking, and planning. We then quantified the frequency of these keywords within the AIME25 responses generated by

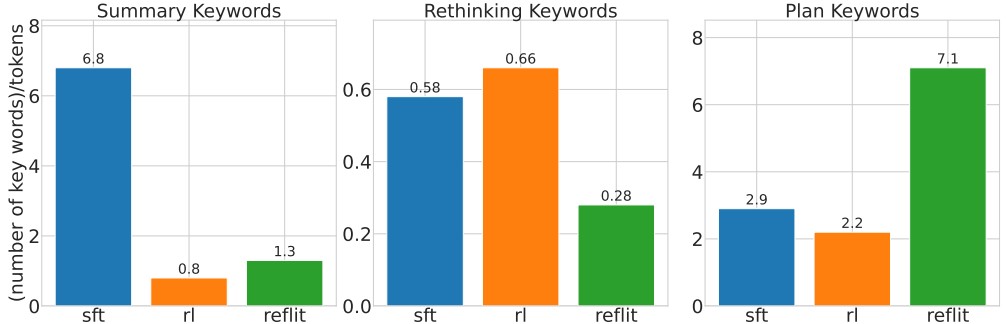

Figure 8: Normalized Keyword Counts for RL, SFT, and ReLIFT Models on AIME25

each model. To ensure a fair comparison across varying response lengths, all keyword counts are normalized by the total number of tokens. As shown in Figure 8, our analysis reveals several key differences in how the models approach problem-solving.

**Summarizing:** The SFT model exhibits a notably higher usage of summary keywords, suggesting it often prioritizes synthesizing and restating information. The ReLIFT model uses slightly more summary keywords than the RL model, but both are far behind SFT.

**Rethinking:** In contrast, the ReLIFT model uses significantly fewer keywords related to rethinking or re-evaluation. This finding suggests a higher level of efficiency and confidence in its initial approach, requiring less internal correction or revision during the problem-solving process.

**Planning:** The most striking difference is in the use of planning keywords. The ReLIFT model demonstrates a substantial increase in these keywords, indicating a more robust and sophisticated approach to strategic planning. This suggests that ReLIFT is not just more efficient but also more capable of developing a logical and powerful problem-solving trajectory.

# F  CASE STUDY

**AIME25 I Problem 3:** The first step to solve this problem is to find three possible combinations $(6, 2, 1)$, $(5, 3, 1)$ and $(4, 3, 2)$. The RL response fails to include one possible combination, while the SFT response requires excessive analysis to obtain the three combinations. In contrast, the ReLIFT response is more concise and obtains all three combinations.

---

**AIME25 I Problem 3**

**Question:** The 9 members of a baseball team went to an ice cream parlor after their game. Each player had a singlescoop cone of chocolate, vanilla, or strawberry ice cream. At least one player chose each flavor, and the number of players who chose chocolate was greater than the number of players who chose vanilla, which was greater than the number of players who chose strawberry. Let $N$ be the number of different assignments of flavors to players that meet these conditions. Find the remainder when $N$ is divided by 1000.

---

**ReLIFT (Correctness: True; Length: 1819 tokens)**

$\cdots$ Let's denote: - $c$ as the number of players who chose chocolate. - $v$ as the number of players who chose vanilla. - $s$ as the number of players who chose strawberry. The conditions given are: 1. $c + v + s = 9$ 2. $(c > v > s)$ 3. Each flavor is chosen by at least one player, so $c \geq 1$, $v \geq 1$, and $s \geq 1$. ### Step 2: Determine Possible Values for $c, v, s$ Since $c, v, s \geq 1$ and $c + v + s = 9$, we can start by considering possible values for $c, v, s$ that satisfy $(c > v > s)$ and all being positive integers. ### Step 3: Find Possible Combinations of $c, v, s$ We need to find all the integer solutions to the equation $c + v + s = 9$ where $(c > v > s \geq 1)$. Let's list possible values for $c, v, s$: First, let's note that $s$ must be at least 1 and since $(v > s)$, $v$ must be at least 2, and since $(c > v)$, $c$ must be at least 3. Also, since $c + v + s = 9$, the maximum value for $c$ can be determined by assuming the minimum values for $v$ and $s$. If $s = 1$ and $v = 2$, then $c = 6$. If $s = 1$ and $v = 3$, then $c = 5$. If $s = 2$ and $v = 3$, then $c = 4$. So possible values for $c$ can be 4, 5, 6. Let's list all possible combinations: - If $c = 6$, then $v + s = 3$ and $(v > s \geq 1)$. Possible pairs are $(v, s) = (2, 1)$. - If $c = 5$, then $v + s = 4$ and $(v > s \geq 1)$. Possible pairs are $((v, s) = (3, 1))$ or $((2, 2))$, but since $(v > s)$, only $((v, s) = (3, 1))$. - If $c = 4$, then $v + s = 5$ and $(v > s \geq 1)$. Possible pairs are $(v, s) = (3, 2)$. So the possible combinations are: $(6, 2, 1)$, $(5, 3, 1)$ and $(4, 3, 2)$ $\cdots$ Final Answer The final answer is $\boxed{016}$.

**RL (Correctness: False; Length: 1269 tokens)**

$\cdots$ Let's denote: - $C$ as the number of players who chose chocolate, - $V$ as the number of players who chose vanilla, - $S$ as the number of players who chose strawberry. We need to satisfy the following conditions: 1. $C + V + S = 9$, 2. $C > V > S \geq 1$. First, let's find all possible combinations of $(C, V, S)$ that satisfy these conditions. We'll iterate through possible values to find all valid triples $(C, V, S)$ and then calculate the number of different assignments for each valid triple. Since $C > V > S \geq 1$ and $C + V + S = 9$, the possible values for $S$ can range from 1 to 2 (since if $S \geq 3$, then $V \geq 4$ and $C \geq 5$, which would make $C + V + S \geq 12$, which is not possible). Let's list out the possible triples: 1. If $S = 1$: - $C + V = 8$ and $C > V > 1$. Possible pairs for $(C, V)$ are $(6, 2)$ and $(5, 3)$. 2. If $S = 2$: - $C + V = 7$ and $C > V > 2$. The only possible pair for $(C, V)$ is $(5, 2)$, but this does not satisfy $V > S$ since $V = 2 = S$. So, $S = 2$ does not yield any valid pairs. Thus, the valid triples are $(6, 2, 1)$ and $(5, 3, 1)$ $\cdots$ Thus, the answer is $\boxed{756}$.

**SFT (Correctness: True; Length: 6636 tokens)**

··· We have 9 players, each choosing one of three flavors: C, V, S. Each flavor must be chosen by at least one player. So, we can't have all 9 choosing C, or only Cs and Vs, etc. Also, the number of Cs > number of Vs > number of Ss. So, we need to count the number of ordered triples (c, v, s) where c + v + s = 9, c ≥ 1, v ≥ 1, s ≥ 1, and c > v > s. Then, for each such triple, we need to compute the number of ways to assign the flavors to the players, which would be 9 choose c, then choose which c players get chocolate, then from the remaining 9 - c players, choose v to get vanilla, and the remaining s get strawberry. So, for each valid (c, v, s), the number of assignments is C(9, c) * C(9 - c, v) * C(9 - c - v, s). But since the players are distinct, once we choose the c players for chocolate, the v from the remaining for vanilla, etc., that's the standard multinomial coefficient. So, for each valid triple (c, v, s), the number is 9! / (c! v! s!). Then, we sum over all valid triples. So, first step: find all triples (c, v, s) such that c + v + s = 9, c ≥ 1, v ≥ 1, s ≥ 1, and c > v > s. Then, for each such triple, compute the multinomial coefficient 9! / (c! v! s!) and sum them all up. Then take that total modulo 1000. So, first, let's list all possible triples (c, v, s) where c > v > s ≥ 1, and c + v + s = 9. Let me start by considering possible values of s. Since s is the smallest, and s ≥ 1, so s can be 1, 2, 3, 4. Let's check: If s = 1, then we need v > 1, so v ≥ 2, and c > v. Also, c + v + 1 = 9 => c + v = 8. Since c > v ≥ 2, and c + v = 8. So possible (v, c): v can be 2, 3, 4, 5, 6 (since c has to be greater than v, and v must be at least 2, and c = 8 - v must be greater than v). Let's check: v = 2: c = 6. So (6, 2, 1). Check c > v > s: 6 > 2 > 1. Yes. v = 3: c = 5. (5, 3, 1). 5 > 3 > 1. Good. v = 4: c = 4. But c must be greater than v, so c = 4 is not > v = 4. So invalid. v = 5: c = 3. But then c = 3, which is not > v = 5. So invalid. v = 6: c = 2. Similarly, invalid. So for s = 1, possible triples are (6,2,1), (5,3,1). Next, s = 2. Then s = 2. So c + v + 2 = 9 => c + v = 7. With c > v > 2. So v must be at least 3, and c = 7 - v > v. Let's check possible v: v = 3: c = 4. (4, 3, 2). 4 > 3 > 2. Valid. v = 4: c = 3. But c = 3 is not > v = 4. Invalid. v = 5: c = 2. Also invalid. So only (4, 3, 2) for s = 2. s = 3: Then c + v + 3 = 9 => c + v = 6. With c > v > 3. So v must be at least 4, c = 6 - v > v. Let's check: v = 4: c = 2. But c = 2 is not > v = 4. Invalid. v = 5: c = 1. Also invalid. So no solutions here. s = 4: Then c + v + 4 = 9 => c + v = 5. With c > v > 4. But v must be at least 5, which would make c = 0, which is invalid. So no solutions here. So the possible triples are: For s = 1: (6,2,1), (5,3,1) For s = 2: (4,3,2) Total of 3 triples. Wait, let me check again. For s = 1: c + v = 8. c > v ≥ 2. So possible pairs: v = 2, c = 6 v = 3, c = 5 v = 4, c = 4 (invalid since c must be > v) v = 5, c = 3 (invalid) v = 6, c = 2 (invalid). So yes, only (6,2,1), (5,3,1) For s = 2: c + v = 7. c > v ≥ 3. So v can be 3, c = 4 v = 4, c = 3 (invalid) v = 5, c = 2 (invalid). So only (4,3,2) s = 3: c + v = 6. c > v ≥ 4. v = 4, c = 2 (invalid). So no. s = 4: same as above. So total three triples: (6,2,1), (5,3,1), (4,3,2) ··· Thus, the remainder when $N$ is divided by 1000 is $\boxed{16}$.

## G  MORE OOD RESULTS

We evaluate four new OOD benchmarks: ARC-Challenge (ARC-c) Clark et al. (2018), GPQA-Diamond (GPQA) Rein et al. (2024), MMLU-Redux (Redux) Gema et al. (2024), and HumanEval Chen et al. (2021), in order to comprehensively test the generalization ability of our method. Among them, ARC-Challenge and GPQA-Diamond focus on reasoning ability in the STEM fields, HumanEval focuses on coding, and MMLU-Redux is a carefully annotated version of MMLU, covering questions from a wider range of domains. The results from these evaluations are summarized in the table below.

Table 5: Overall performance on four diverse benchmarks (ARC-c, GPQA, Redux, Humaneval) based on different models. Bold and underline indicate the best and second-best results, respectively.

| Models | ARC-c | | GPQA | | Redux | | Humaneval |
|---|---|---|---|---|---|---|---|
| | *Acc* | *Len* | *Acc* | *Len* | *Acc* | *Len* | *Acc* |
| **Qwen-Math** | 0.503 | 504 | 0.264 | 1110 | 0.474 | 555 | 0.518 |
| **Qwen-Math-Instruct** | 0.657 | 2881 | 0.228 | 5071 | 0.481 | 3628 | 0.435 |
| **RL w/ SFT Loss** | 0.635 | 1814 | 0.274 | 5942 | 0.591 | 2514 | 0.604 |
| **SFT** | 0.750 | 1515 | 0.265 | 6547 | 0.601 | 2479 | 0.553 |
| **RL** | 0.808 | 639 | 0.401 | 1881 | 0.657 | 863 | 0.581 |
| **LUFFY** | 0.804 | 1168 | **0.441** | 3285 | 0.652 | 1516 | 0.572 |
| **SFT then RL(v1)** | 0.723 | 879 | 0.289 | 3779 | 0.562 | 1217 | 0.548 |
| **SFT then RL(v2)** | 0.728 | 980 | 0.339 | 4490 | 0.620 | 1501 | 0.627 |
| **ReLIFT** | **0.816** | 1321 | 0.431 | 3177 | **0.670** | 1511 | **0.643** |

ReLIFT performs strongly across these benchmarks, achieving the highest average score and ranking either first or second on every single one. This underscores the robust generalization ability enabled by ReLIFT's integration of RL and fine-tuning, regardless of whether the tasks involve code, STEM, or other domains. Notably, while RL excels on ARC-Challenge and MMLU-Redux due to its strong OOD generalization—outperforming other methods combining RL and SFT—ReLIFT still manages to surpass it.

## H ABLATION STUDY ON BUFFER SIZE AND DIFFICULTY THRESHOLD

We conduct a detailed ablation study focusing on two key hyperparameters of ReLIFT: the FT buffer threshold ($M$), which controls the maximum size of the fine-tuning buffer, and the difficulty threshold ($Q$). The difficulty threshold $Q$ acts as the gating function, selectively determining which samples—specifically, those with an accuracy less than or equal to $Q$ (i.e., incorrect/uncertain samples)—are retained for SFT. This evaluation was performed over 400 training steps on the five math benchmarks, with the results summarized in the table 6.

Table 6: Ablation study on buffer size and difficulty threshold

| $Q$ | $M = 32$ | $M = 64$ | $M = 128$ | $M = 256$ |
|---|---|---|---|---|
| 0 | 0.411 | 0.490 | 0.478 | 0.459 |
| 1/8 | 0.405 | 0.404 | 0.493 | 0.471 |
| 1/4 | 0.392 | 0.403 | 0.467 | 0.491 |

We observe that extreme settings for the key hyperparameters—the difficulty threshold $Q$ and the buffer size $M$—led to suboptimal results. Specifically, when the buffer size $M$ is overly small (e.g., $M = 32$) or when a relaxed filtering condition $Q$ is combined with a small $M$ (e.g., **Q** = 1/4 with **M** = 64), the model's performance significantly degrades (down to $0.392$ and $0.403$). This clearly indicates that either an small size buffer or an overly relaxed filtering condition results in too frequent SFT updates, failing to provide the necessary stability during RL. The ablation results reveal the existence of several robust parameter configurations that yield peak performance (approximately $0.49 \pm 0.003$):($M = 64, Q = 0$), ($M = 128, Q = 1/8$) and ($M = 256, Q = 1/4$). All these settings effectively balance data selection and fine-tuning capacity. However, when considering implementation efficiency, the $Q = 0$ condition requires the minimum amount of demonstration data. Therefore, ($M = 64, Q = 0$) remains our most recommended configuration, achieving near-optimal performance while minimizing the complexity of the gating function.

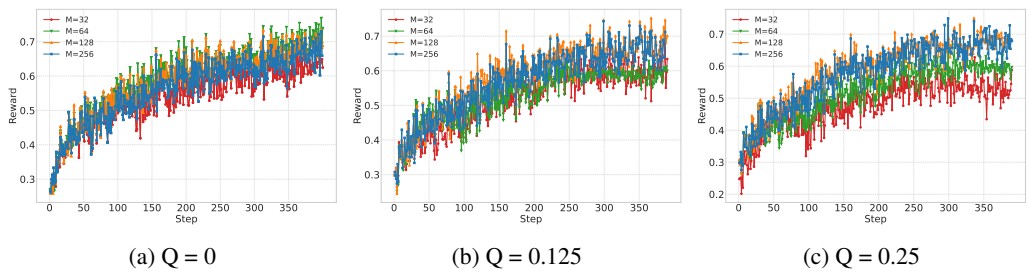

(a) Q = 0        (b) Q = 0.125        (c) Q = 0.25

Figure 9: Training dynamic of rewards during ReLIFT training different Q and M.

The training curves for different combinations of M and Q are shown in Figure 9. It is obvious that extreme settings for $Q$ and $M$ lead to suboptimal results.

