# OpenReview forum: "Learning What Reinforcement Learning Can't: Interleaved Online Fine-Tuning for Hardest Questions"
_ICLR.cc/2026/Conference — ICLR 2026 Poster_

### Official Review · Reviewer_tK9a · 2025-10-21

**Soundness:** 3
**Presentation:** 3
**Contribution:** 2
**Rating:** 4
**Confidence:** 4

**Summary:**

The paper propose to maintain a cache for the hardest questions, which cannot be solved by rollouts. After the cache is full, we will add a sft stage during the rl process. Empircal results show that it can outperfrom sft then rl method.

**Strengths:**

- The paper is easy to read and well-organized.
- The idea is simple but effective.
- Empircal results show the effectiveness of the method.

**Weaknesses:**

- What is the value of this buffer size, $M$? The paper is also missing an analysis experiment (or ablation study) for this. This is important because Figure 4 indicates that the final results seem to be highly dependent on this parameter. And it seems related to the distribution of the datasets.

- The final experimental results show limited improvement over previous methods, and the performance is inconsistent across different datasets. It will be better to show the training curve when compare to luffy rather than the final results since rl is not very stable.

**Questions:**

Do you think this fine-tuning approach is primarily beneficial only when:
- The task or dataset is particularly difficult (e.g., requiring complex multi-step reasoning), or
- The base model is relatively weak and struggles with such questions?

In other words, would the strongest model (e.g., GPT-4-level) still benefit from this approach on datasets?


What will be the results if use rl w/ sft loss instead of sft in ReLIFT?

---

> ### Author Response · Authors · 2025-11-20
> **Author Response**
>
> Dear Reviewer tK9a:
>
> Thank you for your detailed and constructive review. We are pleased that you find our paper 'easy to read' and 'well-organized', and we appreciate you acknowledging that our method is 'simple but effective'. We will carefully address the weaknesses and questions you raised to further strengthen the paper. Our detailed responses follow below.
>
> ### Response to W1
>
> >What is the value of this buffer size? The paper is also missing an analysis experiment (or ablation study) for this. This is important because Figure 4 indicates that the final results seem to be highly dependent on this parameter. And it seems related to the distribution of the datasets.
>
> We sincerely thank your valuable feedback about the limited ablation on the buffer size. As reviewer u4oo and reviewer z9hw all suggest more ablation experiments, we conduct a detailed ablation study on two key hyperparameters of ReLIFT: the fine-tuning buffer threshold (M) and the difficulty threshold (Q). The difficulty threshold $Q$ acts as the gating function, selectively determining which degree of incorrect samples (accuracy<=Q$) are retained for SFT. $M$ controls the maximum size of the fine-tuning buffer. We evaluate the mean performance of various M and Q configurations on the five math benchmarks using the Qwen2.5-Math-7b model, after 400 training steps. The results are shown in the table below. The full results are listed in Appendix H.
>
> | Q    | M=32  | M=64  | M=128 | M=256 |
> |------|-------|-------|-------|-------|
> | 0    | 0.411 | 0.490 | 0.478 | 0.459 |
> | 1/8  | 0.405 | 0.404 | 0.493 | 0.471 |
> | 1/4  | 0.392 | 0.403 | 0.467 | 0.491 |
>
> * We observe that extreme settings for $Q$ and $M$ lead to suboptimal results. Specifically, when the buffer size $M$ is overly small (e.g., $M=32$) or when a relaxed filtering condition $Q$ is combined with a small $M$ (e.g., $Q=1/4$ with $M=64$), the model's performance significantly degrades (down to $0.392$ and $0.403$). This clearly indicates that either an small size buffer or an overly relaxed filtering condition results in too frequent SFT updates, failing to provide the necessary stability during RL.
>
> * The ablation results reveal the existence of several robust parameter configurations that yield peak performance (approximately $0.49\pm0.003$):$(M=64, Q=0)$, $(M=128, Q=1/8)$ and $(M=256, Q=1/4)$. This demonstrates the robustness of our method to the parameters $Q$ and $M$, with multiple suitable hyperparameter choices available.
>
> * The configuration $(M=64, Q=0)$ is still our experiment setting because the $Q=0$ condition demands the minimum amount of demonstration data, enhancing implementation efficiency. Furthermore, as shown in the ablation study (Figure 4), utilizing SFT samples from less difficult questions leads to a notable increase in token length by overriding the model's concise response patterns for simpler problems. Thus, $(M=64, Q=0)$ successfully achieves near-optimal performance while minimizing the complexity of the gating function and mitigating undesirable token length expansion.

---

> > ### Author Response · Authors · 2025-11-20
> > **Author Response**
> >
> > ### Response to W2
> >
> > >The final experimental results show limited improvement over previous methods, and the performance is inconsistent across different datasets. It will be better to show the training curve when compare to luffy rather than the final results since rl is not very stable.
> >
> > We appreciate your concern regarding the limited improvement. To thoroughly address this, we expand our evaluation to include four new challenging OOD benchmarks: ARC-Challenge (**ARC-c**) [1], GPQA-Diamond (**GPQA**) [2], MMLU-Redux (**Redux**) [3], and HumanEval [4]. These benchmarks test a wider range of capabilities beyond mathematical reasoning, including STEM-focused reasoning, general knowledge, and code generation. The results, presented below, show that ReLIFT achieves state-of-the-art or near-state-of-the-art performance across all new benchmarks, demonstrating significantly stronger and more consistent generalization than all baselines. ReLIFT achieves the highest accuracy on ARC-c, MMLU-Redux, and HumanEval, and is highly competitive on GPQA. Notably, while pure RL excels on ARC-Challenge and MMLU-Redux due to its strong OOD generalization—outperforming other SFT+RL methods—ReLIFT still manages to surpass it.
> >
> > | Models | ARC-c | | GPQA | | Redux | | Humaneval |
> > | :--- | :---: | :---: | :---: | :---: | :---: | :---: | :---: |
> > | | **Acc** | **Len** | **Acc** | **Len** | **Acc** | **len** | **Acc** |
> > | qwen2.5-math | 0.503 | 504 | 0.264 | 1110 | 0.474 | 555 | 0.518 |
> > | qwen2.5-math-instruct | 0.657 | 2881 | 0.228 | 5071 | 0.481 | 3628 | 0.435 |
> > | RL w/ SFT Loss | 0.635 | 1814 | 0.274 | 5942 | 0.591 | 2514 | 0.604 |
> > | SFT | 0.750 | 1515 | 0.265 | 6547 | 0.601 | 2479 | 0.553 |
> > | RL | 0.808 | 639 | 0.401 | 1881 | 0.657 | 863 | 0.581 |
> > | LUFFY | 0.804 | 1168 | **0.441** | 3285 | 0.652 | 1516 | 0.572 |
> > | SFT then RL(v1) | 0.723 | 879 | 0.289 | 3779 | 0.562 | 1217 | 0.548 |
> > | SFT then RL(v2) | 0.728 | 980 | 0.339 | 4490 | 0.620 | 1501 | 0.627 |
> > | ReLIFT | **0.816** | 1321 | 0.431 | 3177 | **0.670** | 1511 | **0.643** |
> >
> > Meanwhile, we agree with you that training stability is a critical consideration for RL. To address this, we have included comparative plots of the training curves for LUFFY and ReLIFT in Appendix I wherein ReLIFT demonstrates superior and stable performance, whereas LUFFY exhibits a mid-training collapse phenomenon. LUFFY's model exhibits a sharp initial increase in response length by directly fitting all available off-policy samples, whereas ReLIFT's length increases incrementally. A similar pattern emerges with entropy: LUFFY's spikes at the beginning due to this direct fitting strategy but then gradually declines, indicating diminishing exploration capability. ReLIFT, in contrast, maintains a stable entropy level of approximately 0.2, indicating steady exploration throughout the training process.
> >
> > Finally, we wish to re-emphasize that ReLIFT's advantages are not limited to accuracy. It is also a more efficient paradigm. As shown in Table 2 in paper, ReLIFT achieves its superior performance using significantly less training time and far less demonstration data compared to strong baselines combining SFT and RL. Furthermore, its conceptual design, which adaptively targets emerging model weaknesses with online fine-tuning, is a simple and effective strategy that can be easily adopted or extended by future work.
> >
> > We believe these new OOD results, combined with the evidence for ReLIFT's training stability and superior resource efficiency, significantly strengthen our paper's contribution.

---

> > > ### Author Response · Authors · 2025-11-20
> > > **Author Response**
> > >
> > > ### Response to Q1
> > >
> > > >Would the strongest model (e.g., GPT-4-level) still benefit from this approach on datasets?
> > >
> > > We concur with your hypothesis that the benefits of ReLIFT are indeed most pronounced on highly difficult tasks and for base models whose capabilities are not yet saturated on those tasks. Our primary analysis in Section 2.1 (Figure 1) was designed to demonstrate this precise point. We found a clear complementarity: RL excels at optimizing performance on problems within the model's existing capabilities ('Easy'/'Medium'), but struggles to impart new reasoning patterns for problems beyond its capabilities ('Hardest,' where $acc(q)=0$). Conversely, SFT is essential for teaching these 'Hardest' problems but can degrade performance on simpler ones.
> > > ReLIFT is explicitly designed to leverage this dynamic, using RL for general optimization while surgically interleaving SFT only for the 'Hardest' problems that RL identifies as unsolvable.
> > >
> > > Regarding SOTA models (e.g., GPT-4 level), we hypothesize that ReLIFT remains beneficial, although its role would shift. Any model, regardless of its strength, possesses a 'knowledge boundary' and will encounter 'Hardest' problems. The core mechanism of ReLIFT is adaptive: it dynamically identifies these failures (where $acc(q)=0$) during rollout to trigger fine-tuning. For a SOTA model, the frequency of such failures would be significantly lower, and ReLIFT would automatically reduce the frequency of interleaved fine-tuning steps. However, when such a failure occurs, our findings suggest pure RL would still be insufficient to bridge this knowledge gap. ReLIFT provides a targeted, resource-efficient mechanism to 'patch' these specific model weaknesses using high-quality demonstrations (e.g., from human experts) without risking the performance degradation on simpler problems that a full SFT might cause.
> > >
> > > In summary, while ReLIFT's magnitude of improvement is greatest for weaker models, its adaptive mechanism may provide a valuable and efficient paradigm for targeted knowledge injection, remaining relevant even for SOTA models to correct specific, emergent failures.
> > >
> > >
> > > ### Response to Q2
> > > >What will be the results if use rl w/ sft loss instead of sft in ReLIFT?
> > >
> > > We add this baseline of using RL w/ SFT instead of pure SFT in ReLIFT. The specific implementation is that whenever the stored fine-tuning buffer size exceeds $M$, the next RL step will use the SFT loss of the buffer samples. The specific results are shown in the table below.
> > >
> > > | Models | MATH (Acc) | Olympiad (Acc) | AIME24 (Acc) | AMC (Acc) | AIME25 (Acc) | MMLU-Pro (Acc) | **Overall (Acc)** |
> > > | :--- | :---: | :---: | :---: | :---: | :---: | :---: | :---: |
> > > | ReLIFT(rl w/ sft) | 0.864 | 0.523 | 0.261 | 0.602 | 0.192 | 0.502 | 0.491 |
> > >
> > >
> > > | Models | MATH (Len) | Olympiad (Len) | AIME24 (Len) | AMC (Len) | AIME25 (Len) | MMLU-Pro (Len) | **Overall (Len)** |
> > > | :--- | :---: | :---: | :---: | :---: | :---: | :---: | :---: |
> > > |  ReLIFT(rl w/ sft) | 1565.66 | 3214.62 | 5113.03 | 3234.11 | 4888.74 | 2220.61 | 3372.79 |
> > >
> > > The model demonstrates commendable performance; however, it does not match that of ReLIFT. We hypothesize that this performance gap stems from the conflated optimization objectives, which result in an ambiguous direction for model updates. This issue is likely exacerbated by our current use of a simple 1:1 ratio for the loss components. It is anticipated that a more meticulous hyperparameter tuning, particularly on the loss weights, could yield superior results.
> > >
> > > Please let us know if we have addressed your concerns. We are more than delighted to have further discussions and improve our manuscript. If our response has addressed your concerns, we would be grateful if you could re-evaluate our work.
> > >
> > > [1] Clark, P. et al. Think you have Solved Question Answering? Try ARC, the AI2 Reasoning Challenge. *arXiv:1803.05457v1*, 2018.
> > >
> > > [2] Rein, D. et al. Gpqa: A graduate-level google-proof q\&a benchmark. *First Conference on Language Modeling*, 2024.
> > >
> > > [3] Gema, A.P. et al. Are We Done with MMLU?. *arXiv:2406.04127*, 2024.
> > >
> > > [4] Chen, M. et al. Evaluating Large Language Models Trained on Code. *arXiv:2107.03374*, 2021.

---

> > > > ### Comment · Reviewer_tK9a · 2025-11-20
> > > >
> > > > Thank you for your response and I have updated my score to 6

---

> > > > > ### Author Response · Authors · 2025-11-20
> > > > >
> > > > > Thank you for your suggestions and for the rating increase.

---

### Official Review · Reviewer_H2Ni · 2025-10-28

**Soundness:** 3
**Presentation:** 3
**Contribution:** 3
**Rating:** 8
**Confidence:** 4

**Summary:**

The paper examines the differences between how LLMs learn to solve reasoning problems across various difficulties when using SFT or RL. Informed by this analysis, the paper proposes selectively using demonstrations for intermittent SFT stages during online RL training when faced with the "hardest" problems (i.e., those for which no solution in the GRPO group is correct). The primary results show that this is, for the most part, more effective than appropriate baselines at greater compute efficiency.

**Strengths:**

The paper presents a method that is grounded in a detailed and revealing analysis about the differing impact of RL and SFT. The core results show that the proposed method yields benefits in-domain (math benchmarks) and on a single out-of-domain benchmark (MMLU-Pro). The analysis of training dynamics and the ablation studies effectively reveal how the proposed method mitigates the limitations of conducting SFT or RL alone, or as distinct training stages.

**Weaknesses:**

ReLIFT is considerably less effective on LLaMA-3.1-8B than on Qwen models. It would be useful to include the full set of baselines for a model outside of the Qwen model family, so as to ensure the generality of the method. Currently, only SFT or RL alone, in addition to the instruct variant, are used as baselines for the Llama model.

Using a fixed group size of 8 for all experiments means that it is unclear whether the problem difficulty classes assigned during online RL hold for a larger group size and how this impacts the effectiveness of ReLIFT and baselines. Using at least one more group size (say 16) for the core experiments would be insightful.

In practice, all demonstrations are collected offline before finetuning, which does not fit one of the core motivations for ReLIFT: only collecting demonstrations where necessary. However, collecting demonstrations on-the-fly would add to the complexity of the training pipeline and introduce latency. Discussion of this limitation would be useful.

(Nit) Using the same underline for the second and third best result reduces clarity.

It is unclear whether the entropy regularisation term used in ReLIFT is also used for SFT baselines. This is an important baseline variant that is missing if I am not mistaken.

**Questions:**

Have you tried the SFT baselines with entropy regularisation?

It is not clear if the stable response length for the hardest problems is related to exploration or an artefact of prior training stages. For example, it could be because the response length for these problems is already at the maximum that was seen in training prior to RL and that the demonstrations used for SFT are longer. Could you clarify how response length in the demonstration data compares to generated response lengths from the trained model?

How do the results change if the group size is doubled?

---

> ### Author Response · Authors · 2025-11-20
> **Author Response**
>
> Dear Reviewer H2Ni:
>
> We sincerely appreciate your thoughtful, positive, and constructive review. We are especially grateful for your comments highlighting our "detailed and revealing analysis" of RL and SFT. It is also wonderful to hear that you find our algorithm to "mitigate the limitations of conducting SFT or RL alone." Now, let us address your questions one by one.
>
> ### Response to W1
>
> >Lack set of baselines for a model outside of the Qwen model family
>
> We appreciate your observation regarding the limited baselines for the Llama model. For the Llama3.1-8B model, we test a comprehensive set of baselines, including two versions of SFT followed by RL, LUFFY, and RL with SFT loss.
>
> Both LUFFY and RL with SFT loss consistently fail to effectively elicit Llama3.1-8B's reasoning capabilities. Specifically, their training process exhibit rapid overfitting to off-policy samples, causing the output length to quickly reach the maximum limit. While producing lengthy responses, the models don't generate the correct results. Due to the output truncation, the model never manage to output the complete, correct answer and consequently receive no reward. The SFT-then-RL variants are configured as follows: version 1 is trained for 200 RL steps from the SFT checkpoint in the baseline, while version 2 use nearly 3k demonstration data (aligning with the quantity used by ReLIFT) for three epochs of SFT, followed by 200 RL steps. The final test results are presented in the table below.
>
> | Models | AIME24 (Acc) | AIME25 (Acc) | AMC (Acc) | MATH500 (Acc) | Olympiad (Acc) | MMLU-Pro (Acc) | **Overall (Acc)** |
> | :--- | :---: | :---: | :---: | :---: | :---: | :---: | :---: |
> | SFT then RL v2| 0.010 | 0.004 | 0.099 | 0.328 | 0.098 | 0.328 | 0.145 |
> | SFT then RL v1| 0.008 | 0.004 | 0.140 | 0.386 | 0.113 | 0.383 | 0.174 |
> | ReLIFT | 0.013 | 0.002 | 0.119 | 0.352 | 0.110 | 0.442 | 0.173 |
>
> | Models | AIME24 (Len) | AIME25 (Len) | AMC (Len) | MATH500 (Len) | Olympiad (Len) | MMLU-Pro (Len) | **Overall (Len)** |
> | :--- | :---: | :---: | :---: | :---: | :---: | :---: | :---: |
> | SFT then RL v2| 1945 | 1958 | 1871 | 1642 | 1843 | 1427 | 1781 |
> | SFT then RL v1| 1901 | 1851 | 1842 | 1621 | 1832 | 1207 | 1709 |
> | ReLIFT | 1236 | 1272 | 1048 | 741 | 1050 | 650 | 1000 |
>
> The SFT-then-RL v1 model only outperforms ReLIFT in the domain of mathematics; however, ReLIFT's MMLU-Pro score is significantly higher. This outcome indicates that continuing with RL after SFT on a large amount of mathematical data does not fully recover the model's OOD generalization capabilities. Furthermore, the SFT-then-RL v2 model, despite using the same amount of demonstration data as ReLIFT, performs worse than ReLIFT in both mathematical and OOD tasks. ReLIFT is the sole method that consistently maintains strong performance across both mathematical and OOD tasks.
>
> ### Response to W2
>
> >Using at least one more group size (say 16) for the core experiments would be insightful.
>
> We appreciate your observation regarding the limited training group size. We evaluate the performance of RL and ReLIFT with a Group size of 16. To maintain a computational budget roughly equivalent to the Group=8 setting, the models are trained for 300 steps. The resulting data is presented in the table below.
>
> | Models | AIME24 (Acc) | AIME25 (Acc) | AMC (Acc) | MATH500 (Acc) | Olympiad (Acc) | MMLU-Pro (Acc) | **Overall (Acc)** |
> | :--- | :---: | :---: | :---: | :---: | :---: | :---: | :---: |
> | RL | 0.241 | 0.162 | 0.589 | 0.832 | 0.475 | 0.482 | 0.460 |
> | ReLIFT | 0.246 | 0.171 | 0.613 | 0.872 | 0.502 | 0.508 | 0.481 |
>
> | Models | AIME24 (Len) | AIME25 (Len) | AMC (Len) | MATH500 (Len) | Olympiad (Len) | MMLU-Pro (Len) | **Overall (Len)** |
> | :--- | :---: | :---: | :---: | :---: | :---: | :---: | :---: |
> | RL | 2926 | 2494 | 1705 | 1004 | 1737 | 1005 | 1741 |
> | ReLIFT | 3804 | 3743 | 3174 | 2492 | 3063 | 2295 | 3064 |
>
> The results clearly indicate that ReLIFT remains effective even at Group=16. This is because, despite the increased number of rollouts, the method consistently encounters samples that are beyond the model's current capability (where accuracy is 0). Crucially, these challenging samples continue to provide valuable and effective guidance for training.

---

> > ### Author Response · Authors · 2025-11-20
> > **Author Response**
> >
> > ### Response to W3
> >
> > >In practice, all demonstrations are collected offline before finetuning, which does not fit one of the core motivations for ReLIFT: only collecting demonstrations where necessary. However, collecting demonstrations on-the-fly would add to the complexity of the training pipeline and introduce latency. Discussion of this limitation would be useful.
> >
> > You are correct that our current experimental setup, where all fine-tuning demonstration data is collected offline prior to training, introduces a practical divergence from ReLIFT's core motivation of collecting high-quality demonstrations online only when necessary.
> > We appreciate this critical observation and provide the following explanation and discussion regarding this limitation and the technical feasibility of the online approach. In our revised PDF (Section 6), we have added discussion of this limitation.
> >
> > * The current system design, which utilizes an offline collection method for high-quality demonstrations, is a necessity driven by resource constraints. Specifically, we are unable to implement a fully "online collection" pipeline due to a lack of stable human annotation resources. Generating high-quality demonstrations in real-time and at a large scale during training demands substantial computational and financial support. Therefore, to ensure the experiment's controllability, reproducibility, and cost-effectiveness, we opted to prepare all potential "high-quality CoT solutions" in advance using the offline collection method, a constraint that will be explicitly clarified in the paper's Discussion section.
> >
> > * "Online Collection" of high-quality data, although demonstrated here with offline data, is technically achievable without significantly impacting the overall training time, thanks to Asynchronous Processing. The core idea is that the substantial time taken by RL policy update step is sufficient to mask or cover the latency of collecting new data. While the RL update is running, the system can asynchronously dispatch the identified "hardest" questions (those with 0% rollout accuracy) to either a stronger model API (like DeepSeek-R1) or a human annotation queue. Since an API call and subsequent processing for one or even multiple data points are typically much faster than a model update, this easily fits within the duration of the RL step. Therefore, as long as the rate at which the RL rollout identifies these challenging problems aligns with the rate at which the external source (API or human) generates high-quality solutions, this online collection process can be integrated seamlessly, maintaining the original training speed.
> >
> > We fully agree that, given sufficient resources, implementing a fully "online" ReLIFT system combined with high-quality human annotation is a highly valuable direction for future research. In our revised PDF (Section 6), we have added discussion of this limitation.
> >
> > ### Response to W4
> >
> > >Using the same underline for the second and third best result reduces clarity
> >
> > We have changed the display of the table; the second and third results are indicated with a single underline and a double underline respectively.
> >
> > ### Response to Q1
> >
> > >The lack of SFT baselines with entropy regularisation
> >
> > We add the SFT with entropy regularization baseline, and the results are shown in the table below. The performance is comparable to SFT, but the OOD performance is a little stronger.
> >
> > | Models | AIME24 (Acc) | AIME25 (Acc) | AMC (Acc) | MATH500 (Acc) | Olympiad (Acc) | MMLU-Pro (Acc) | **Overall (Acc)** |
> > | :--- | :---: | :---: | :---: | :---: | :---: | :---: | :---: |
> > | SFT | 0.269 | 0.255 | 0.598 | 0.856 | 0.536 | 0.444 | 0.493 |
> > | SFT w/ Entropy Loss | 0.223 | 0.219 | 0.615 | 0.852 | 0.537 | 0.463 | 0.485 |
> >
> > | Models | AIME24 (Len) | AIME25 (Len) | AMC (Len) | MATH500 (Len) | Olympiad (Len) | MMLU-Pro (Len) | **Overall (Len)** |
> > | :--- | :---: | :---: | :---: | :---: | :---: | :---: | :---: |
> > | SFT | 7344 | 7059 | 5675 | 3503 | 5662 | 3956 | 5533 |
> > | SFT w/ Entropy Loss | 7520 | 7742 | 5562 | 3918 | 5728 | 3879 | 5725 |

---

> > > ### Author Response · Authors · 2025-11-20
> > > **Author Response**
> > >
> > > ### Response to Q2
> > >
> > > > It is not clear if the stable response length for the hardest problems is related to exploration or an artefact of prior training stages. Clarify how response length in the demonstration data compares to generated response lengths from the trained model?
> > >
> > > The concern that ReLIFT's shorter average output length is is no related to exploration but an artefact of prior training stages is s a crucial point. To address this empirically, we used a 939-sample non-overlapping test set from OpenR1 dataset, filtering for the 190 hardest problems (0% accuracy for our trained ReLIFT). We then compare the lengths of ReLIFT's generations (response_len) to the target solutions (target_length) for these specific problems. The statistical comparison is as follows:
> > >
> > > | Statistic | response_len | target_length |
> > > |-----------|--------------|---------------|
> > > | mean      | 4,569.02     | 6,811.79      |
> > > | std       | 1,853.76     | 2,180.59      |
> > > | min       | 725.38       | 435.00        |
> > > | 25th percentile | 3,169.47 | 5,957.00      |
> > > | 50th percentile (median) | 4,503.69 | 8,049.50 |
> > > | 75th percentile | 5,892.63 | 8,124.75      |
> > > | 90th percentile | 7,089.56 | 8,165.10      |
> > > | 95th percentile | 7,809.19 | 8,183.00      |
> > >
> > > The table clearly shows that ReLIFT is not simply mimicking the training data's length:
> > >
> > > * ReLIFT's mean response (4,569.02) is significantly shorter than the target's (6,811.79).
> > >
> > > * The target lengths are heavily clustered at the high end (median 8,049.50), while ReLIFT's outputs are far more spread out and varied.
> > >
> > > This demonstrates that the model is learning the reasoning process itself, not overfitting to the specific length artifact of the demonstration data.
> > >
> > > We hope our responses answer your questions, and again, thanks for your advice to improve our paper!

---

> > > > ### Comment · Reviewer_H2Ni · 2025-11-21
> > > > **Acknowledging author responses**
> > > >
> > > > Thank you for the detailed responses and for taking on the feedback. Including these additions in the revised manuscript will strengthen the paper's claims.

---

> > > > > ### Author Response · Authors · 2025-11-22
> > > > >
> > > > > We sincerely thank you for your support of our work. Your valuable feedback has significantly contributed to improving our paper's quality!

---

### Official Review · Reviewer_u4oo · 2025-11-01

**Soundness:** 3
**Presentation:** 2
**Contribution:** 2
**Rating:** 6
**Confidence:** 3

**Summary:**

The paper explores the complementarity between Reinforcement Learning (RL/RLVR) and Supervised Fine-Tuning (SFT) for reasoning LLMs, and proposes ReLIFT—an alternating training strategy.

During RL, the method identifies the “hardest” problems (current rollout accuracy acc(q)=0), collects or generates high-quality CoT solutions into a buffer, and periodically performs an SFT step before returning to RL. On five math reasoning benchmarks and one OOD benchmark with Qwen2.5-Math-7B, ReLIFT improves overall accuracy over RLVR and mixed RL+SFT baselines, while producing shorter outputs and requiring less training time and fewer examples.

**Strengths:**

1. The paper clearly defines the GRPO objective and the alternating SFT loss with entropy regularization (α), and the training curves show how reward, length, and entropy evolve over steps, supporting a mechanism of continued exploration and steady gains.

2. The method is easy to follow thanks to the flow diagram (Figure 2), the difficulty stratification, and the explicit buffer trigger condition
（Buffer_ft >= M）which together make reproduction straightforward.

**Weaknesses:**

1. The OOD evaluation relies only on MMLU-Pro and the main experiments focus on math reasoning; please add code, science QA, and multi-step commonsense tasks to test adaptability under different verifiable rewards.

2.  Beyond **acc(q)=0**, please evaluate thresholds based on uncertainty, length anomalies, or self-contradictions in the CoT, and formalize the adaptive SFT trigger as a gating function of reward or entropy; report learning curves under different gating hyperparameters.

3. Although ReLIFT yields shorter outputs on average, please check whether necessary long reasoning is over-penalized on AIME-style problems and provide length–accuracy analyses to identify such cases.

**Questions:**

see weakness.

---

> ### Author Response · Authors · 2025-11-20
> **Author Response**
>
> Dear Reviewer u4oo:
>
> We would like to extend our sincere thanks for your positive and constructive review. We are delighted that you found our analysis of RL ans SFT to be "detailed and revealing". We particularly appreciate you highlighting our algorithm design "easy to follow" and "more effective at greater compute efficiency". We will address your questions in detail below.
>
> ### Response to W1
>
> >No evaluation on code, science QA, and multi-step commonsense tasks
>
> We appreciate your observation regarding the limited scope of our original evaluations. In our revised PDF (Appendix G), we add four new benchmarks: ARC-Challenge (**ARC-c**) [1], GPQA-Diamond (**GPQA**) [2], MMLU-Redux (**Redux**) [3], and HumanEval [4], in order to comprehensively test the generalization ability of our method. Among them, ARC-Challenge and GPQA-Diamond focus on reasoning ability in science QA, HumanEval focuses on coding, and MMLU-Redux is a carefully annotated version of MMLU, covering questions from a wider range of domains, including multi-step commonsense tasks. The results from these evaluations are summarized in the table below.
>
> | Models | ARC-c | | GPQA | | Redux | | Humaneval |
> | :--- | :---: | :---: | :---: | :---: | :---: | :---: | :---: |
> | | **Acc** | **Len** | **Acc** | **Len** | **Acc** | **len** | **Acc** |
> | qwen2.5-math | 0.503 | 504 | 0.264 | 1110 | 0.474 | 555 | 0.518 |
> | qwen2.5-math-instruct | 0.657 | 2881 | 0.228 | 5071 | 0.481 | 3628 | 0.435 |
> | RL w/ SFT Loss | 0.635 | 1814 | 0.274 | 5942 | 0.591 | 2514 | 0.604 |
> | SFT | 0.750 | 1515 | 0.265 | 6547 | 0.601 | 2479 | 0.553 |
> | RL | 0.808 | 639 | 0.401 | 1881 | 0.657 | 863 | 0.581 |
> | LUFFY | 0.804 | 1168 | **0.441** | 3285 | 0.652 | 1516 | 0.572 |
> | SFT then RL(v1) | 0.723 | 879 | 0.289 | 3779 | 0.562 | 1217 | 0.548 |
> | SFT then RL(v2) | 0.728 | 980 | 0.339 | 4490 | 0.620 | 1501 | 0.627 |
> | ReLIFT | **0.816** | 1321 | 0.431 | 3177 | **0.670** | 1511 | **0.643** |
>
> ReLIFT demonstrates robust performance across all benchmarks, achieving the highest average score and ranking first or second in every task. This consistent success highlights its superior generalization ability, stemming from the effective integration of RL and fine-tuning. Notably, while pure RL excels on ARC-Challenge and MMLU-Redux due to its strong OOD generalization—outperforming other SFT+RL methods—ReLIFT still manages to surpass it.

---

> ### Author Response · Authors · 2025-11-20
> **Author Response**
>
> ### Response to W2
>
> >Beyond acc(q)=0, please evaluate thresholds based on uncertainty, length anomalies, or self-contradictions in the CoT, and formalize the adaptive SFT trigger as a gating function of reward or entropy; report learning curves under different gating hyperparameters.
>
> We sincerely appreciate your valuable feedback, echoed by reviewers z9hw and tK9a, regarding the limited ablation study on our gating hyperparameters. In response, we conduct a detailed ablation study focusing on two key hyperparameters of ReLIFT: the fine-tuning buffer threshold ($M$), which controls the maximum size of the fine-tuning buffer, and the difficulty threshold ($Q$). The difficulty threshold $Q$ acts as the gating function, selectively determining which samples—specifically, those with an accuracy less than or equal to $Q$—are retained for SFT. All experiments are performed over 400 training steps on the five math benchmarks, with the average accuracy summarized in the table below and learning curves under different gating hyperparameters provided in Appendix H.
>
> | Q    | M=32  | M=64  | M=128 | M=256 |
> |------|-------|-------|-------|-------|
> | 0    | 0.411 | 0.490 | 0.478 | 0.459 |
> | 1/8  | 0.405 | 0.404 | 0.493 | 0.471 |
> | 1/4  | 0.392 | 0.403 | 0.467 | 0.491 |
>
> * We observe that extreme settings for the difficulty threshold $Q$ and the buffer size $M$—lead to suboptimal results. Specifically, when the buffer size $M$ is overly small (e.g., $M=32$) or when a relaxed filtering condition $Q$ is combined with a small $M$ (e.g., $Q=1/4$ with $M=64$), the model's performance significantly degrades (down to $0.392$ and $0.403$). This clearly indicates that either an small size buffer or an overly relaxed filtering condition results in too frequent SFT updates, failing to provide the necessary stability during RL.
>
> * The ablation results reveal the existence of several robust parameter configurations that yield peak performance (approximately $0.49\pm0.003$):$(M=64, Q=0)$, $(M=128, Q=1/8)$ and $(M=256, Q=1/4)$. This demonstrates the robustness of our method to the parameters $Q$ and $M$, with multiple suitable hyperparameter choices available. The configuration $(M=64, Q=0)$ is our experiment setting because the $Q=0$ condition demands the minimum amount of demonstration data, enhancing implementation efficiency. Furthermore, as shown in the ablation study (Figure 4), utilizing SFT samples from less difficult questions leads to a notable increase in token length by overriding the model's concise response patterns for simpler problems. Thus, $(M=64, Q=0)$ successfully achieves near-optimal performance while minimizing the complexity of the gating function and mitigating undesirable token length expansion.
>
> We acknowledge the highly valuable suggestions to explore more SFT triggering mechanisms, such as those based on uncertainty, length anomalies, or CoT self-contradictions. While these represent promising avenues for future work, we deliberately limit our current investigation to the two most critical and readily interpretable control parameters, $M$ and $Q$, which are directly derived from the core reward signal.
>
> * Uncertainty: Uncertainty-based triggering can be categorized into two forms. For question-level uncertainty (predicting uncertainty), model accuracy is an excellent metric, and our $Q$-based trigger is a simpler implementation aligned with this. For answer-level uncertainty, the trigger is related to the response entropy. While we acknowledge that response entropy constitutes a promising triggering signal, we find it less intuitive than accuracy, and its continuous nature introduces significant complexity in setting and tuning a reliable threshold.
>
> * Length Anomalies and CoT Self-Contradiction: Metrics like "length anomalies" lack a precise, universally applicable formal definition or mathematical formula across different problem domains. Furthermore, incorporating highly complex logic such as CoT self-contradiction analysis would introduce significant engineering overhead and tuning complexity, which runs counter to our goal of providing a simple and robust method.
>
> In summary, the ablation on $Q$ and $M$ successfully demonstrates the efficacy and robustness of a reward-based adaptive gating function. We acknowledge the highly valuable suggestions to explore more SFT triggering mechanisms; however, within the scope of the current paper, and to emphasize the core framework corresponding to our primary motivation, we restrict our discussion to the two most direct triggering parameters, $M$ and $Q$.

---

> > ### Author Response · Authors · 2025-11-20
> > **Author Response**
> >
> > ### Response to W3
> >
> > >Check whether necessary long reasoning is over-penalized on AIME-style problems and provide length–accuracy analyses to identify such cases.
> >
> > We thank you for this crucial question. The concern that ReLIFT's optimization for shorter average length might "over-penalize necessary long reasoning" on complex problems like AIME is a valid and important point.
> >
> > To investigate this empirically, we conduct the precise length-accuracy analysis on AIME25 that you suggested. We categorize questions into bins based on their average response token length from trained ReLIFT model. The results, presented in the table below, show that ReLIFT does not penalize necessary length; rather, it successfully promotes reasoning efficiency and adaptivity, avoiding the failure modes of both pure RL and SFT.
> >
> > | Avg Token Range | RL - Questions | RL - Avg Accuracy (%) | SFT - Questions | SFT - Avg Accuracy (%) | ReLIFT (Ours) - Questions | ReLIFT (Ours) - Avg Accuracy (%) |
> > |-----------------|----------------|-----------------------|-----------------|------------------------|---------------------------|-----------------------------------|
> > | 0-2000          | 13             | 32.93%                | 0               | N/A                    | 1                         | 100.00%                           |
> > | 2001-3000       | 5              | 10.00%                | 0               | N/A                    | 5                         | 70.00%                            |
> > | 3001-4000       | 4              | 0.78%                 | 3               | 93.75%                 | 6                         | 25.52%                            |
> > | 4001-5000       | 3              | 1.04%                 | 3               | 83.33%                 | 6                         | 5.73%                             |
> > | 5000-6000       | 4              | 0.78%                 | 0               | N/A                    | 4                         | 2.34%                             |
> > | 6000-7000       | 1              | 0.00%                 | 3               | 45.83%                 | 5                         | 4.38%                             |
> > | 7000-8000       | 0              | N/A                   | 21              | 5.65%                  | 3                         | 2.00%                             |
> >
> >
> > This analysis yields three key insights that directly address the concern:
> >
> > * SFT's Inefficient, Pathological Verbosity: The SFT baseline is not a good proxy for "necessary long reasoning." It exhibits a strong pathological bias for excessive length, with 21 of 30 questions falling into the 7000-8000 token range, where it achieves a dismal 5.65% accuracy. This indicates SFT learns to be verbose, not necessarily correct, and ReLIFT rightly avoids this inefficient, low-accuracy behavior.
> >
> > * RL's Incapability for Long Reasoning: The pure RL baseline demonstrates the opposite failure. It is incapable of sustained long-form reasoning, clustering at the short end (18 of 30 questions are < 3000 tokens). Its accuracy collapses as length increases (e.g., 0.78% in the 3-4k range), and it fails to generate any responses over 7000 tokens, suggesting it simply gives up on the most complex problems.
> >
> > * ReLIFT's True Adaptivity: ReLIFT demonstrates a balanced and adaptive length distribution, solving the problems of both baselines. Unlike pure RL, ReLIFT is fully capable of generating long-form reasoning. It successfully produces responses across all length bins, including 5 questions in the 6-7k range and 3 in the 7-8k range. It tackles the hard problems that require extensive solutions. Unlike SFT, ReLIFT does not default to being long. For simpler problems, it produces highly accurate, concise answers (e.g., 100% accuracy in the 0-2k bin and 70% in the 2-3k bin).
> >
> > This adaptivity is reinforced by our qualitative analyses. As shown in Appendix F (AIME25 Problem 3), ReLIFT provides a correct answer that is significantly more concise than SFT's "excessive" but also correct solution. Appendix E further shows that ReLIFT's reasoning behavior involves more "planning" and less "rethinking," suggesting it builds a more confident, direct, and efficient reasoning path.
> >
> > In summary, ReLIFT does not over-penalize necessary length. Instead, it learns to be adaptively efficient, avoiding the pathological verbosity of SFT and the capability collapse of RL.
> >
> > We hope that our response could address your concerns, and again, thanks for your advice to improve our paper!
> >
> > [1] Clark, P. et al. Think you have Solved Question Answering? Try ARC, the AI2 Reasoning Challenge. *arXiv:1803.05457v1*, 2018.
> >
> > [2] Rein, D. et al. Gpqa: A graduate-level google-proof q\&a benchmark. *First Conference on Language Modeling*, 2024.
> >
> > [3] Gema, A.P. et al. Are We Done with MMLU?. *arXiv:2406.04127*, 2024.
> >
> > [4] Chen, M. et al. Evaluating Large Language Models Trained on Code. *arXiv:2107.03374*, 2021.

---

### Official Review · Reviewer_z9hW · 2025-11-04

**Soundness:** 3
**Presentation:** 3
**Contribution:** 3
**Rating:** 6
**Confidence:** 4

**Summary:**

Current form of RL for LLMs is insufficient to induce capabilities that exceed the limitations of the base model. This paper targets at how to train reasoning-focused LLMs using RL interleaved with SFT. The paper claims that RL mostly explits what the base model already knows and SFT is better at teaching new and harder behaviors. Though SFT requires a lot of demonstrations and sometimes damage the performance oneasier questions. The paper introduces ReLIFT (Reinforcement Learning Interleaved with Online Fine-Tuning), a training framework that mixes RL and SFT adaptively rather than in fixed stages. The method first do RL on a dataset and collect hardest questions at the same time. When the hard-question buffer is full, it will invoke a SFT on these hard questions, followed by another round of RL training. Experiments show that ReLIFT outperforms pure RL, pure SFT, and prior hybrid/RLVR methods across multiple math benchmarks and an OOD benchmark, achieving higher accuracy with fewer demonstrations and more concise reasoning, suggesting that targeted, online SFT on model's failures is an efficient way to expand a model’s reasoning abilities.

**Strengths:**

* The paper first do analysis on training dynamics of RL vs SFT across different difficulties, showing that RL mainly preserves existing skills while SFT can unlock previously unsolved questions. After that, they propose ReLIFT, which does RL as the primary loop while streaming in SFT steps only on hard questions. The method proposed is gounded on grounded observation, which makes sense.
* In the experiments, the paper includes careful controlled comparisons of pure RL, pure SFT, and multiple hybrid baselines (e.g., RL+SFT loss, LUFFY, ...) on the subset of OpenR1-Math-220k dataset, with Qwen2.5 model. The performance of the model trained with ReLIFT seems to beat all baselines and prior RLvR methods while using less detailed demonstrations.
* The paper is well-structured and easy to follow. The motivation of the proposed method is illustrated clearly in Section 2 and the introduction of the method is detailed. The reproduction of the method is very likely.
* The framework is simple, model-agnostic, and shown to generalize across different model sizes and architectures, it’s likely to influence how future reasoning LLMs combine RL and SFT

**Weaknesses:**

* The proposed method ReLIFT is built upon the assumption that high-quality CoT already exists so that smaller LLMs can benifit (it's actually a form of distillation from large LLMs like DeepSeek-R1). Although the paper also mentions that such high-quality CoT may come from human annotators in line 202, detailed discussion on "human annotation" is missing. Therefore, the scope of this paper seems limited to how to efficiently distill from larger LLMs to improve the performance of smaller LLMs, rather than pushing the performance boundary of exisiting LLMs.
* All main experiments are on five math benchmarks plus a single OOD benchmark (MMLU-Pro). There’s no evaluation on code / STEM or other tasks, which can't fully support the claim that "ReLIFT is a powerful and resource-efficient paradigm for developing capable **reasoning** models."
* “Hardest” questions are defined as those with rollout accuracy acc(q)=0 given N samples. Those questions are the only ones that enter the buffer then for SFT. This binary cut can misclassify borderline questions (e.g., model answers corretly only once) and depends heavily on sampling noise. There’s no ablation on using thresholds like acc(q) ≤ p, or on how many “hardness levels” might be useful.

**Questions:**

* line 300: The paper mentions that temperature is set to 0.6 in all evaluations. However, for OlympiadBench and MATH500, the paper states that "For OlympiadBench and MATH500, we use pass@1 as the evaluation metric". I wonder if this is a reliable evaluation on these two benchmarks, as setting temperature=0.6 indicates that the response from the trained model is not deterministic.

---

> ### Author Response · Authors · 2025-11-20
> **Author Response**
>
> Dear Reviewer z9hW:
>
> Thank you for your careful reading of our paper and for your thoughtful, constructive review. We are very encouraged by your positive feedback and appreciate that you recognize our work as "likely to influence how future reasoning LLMs combine RL and SFT". Let us address your questions one by one.
>
> ### Response to W1
>
> >The proposed method ReLIFT is built upon the assumption that high-quality CoT already exists so that smaller LLMs can benifit (it's actually a form of distillation from large LLMs like DeepSeek-R1). Although the paper also mentions that such high-quality CoT may come from human annotators in line 202, detailed discussion on "human annotation" is missing. Therefore, the scope of this paper seems limited to how to efficiently distill from larger LLMs to improve the performance of smaller LLMs, rather than pushing the performance boundary of exisiting LLMs.
>
> We appreciate your comment and wish to clarify a key distinction. While our experiments use data from a stronger model (DeepSeek-R1) for reproducibility, our method's core innovation is not distillation.
> The central mechanism of ReLIFT—"online identification of hard problems and interleaved SFT"—is source-agnostic. We select DeepSeek-R1 for experimental convenience, but the framework is designed to work just as well with other data sources, such as the human annotators mentioned in the paper. Owing to the low requirement for demonstration data, ReLIFT maintains the feasibility of using human annotation.
>
> Though resource constraints prevent human-in-the-loop experiments, our asynchronous design ensures this is technically achievable. The latency of new data collection (e.g., from a human) is masked by the long duration of RL policy update, allowing data gathering to occur in parallel without slowing training.
>
> We fully agree that, given sufficient resources, implementing a fully "online" ReLIFT system combined with high-quality human annotation is a highly valuable direction for future research. In our revised PDF (Section 6), we have added discussion to clarify our core contributions. This section also discusses the feasibility of integrating human annotations into ReLIFT, a promising direction given the model's minimal data requirements. While resource constraints prevented us from conducting experiments with human-annotated data, we highlight this as a critical area for future research.
>
> ### Response to W2
>
> >All main experiments are on five math benchmarks plus a single OOD benchmark (MMLU-Pro). There’s no evaluation on code / STEM or other tasks, which can't fully support the claim that "ReLIFT is a powerful and resource-efficient paradigm for developing capable reasoning models."
>
> We appreciate your observation regarding the limited scope of our original evaluations. In our revised PDF (Appendix G), we add four new benchmarks: ARC-Challenge (**ARC-c**) [1], GPQA-Diamond (**GPQA**) [2], MMLU-Redux (**Redux**) [3], and HumanEval [4], in order to comprehensively test the generalization ability of our method. Among them, ARC-Challenge and GPQA-Diamond focus on reasoning ability in the STEM fields, HumanEval focuses on coding, and MMLU-Redux is a carefully annotated version of MMLU, covering questions from a wider range of domains. The results from these evaluations are summarized in the table below.
>
> | Models | ARC-c | | GPQA | | Redux | | Humaneval |
> | :--- | :---: | :---: | :---: | :---: | :---: | :---: | :---: |
> | | **Acc** | **Len** | **Acc** | **Len** | **Acc** | **len** | **Acc** |
> | qwen2.5-math | 0.503 | 504 | 0.264 | 1110 | 0.474 | 555 | 0.518 |
> | qwen2.5-math-instruct | 0.657 | 2881 | 0.228 | 5071 | 0.481 | 3628 | 0.435 |
> | RL w/ SFT Loss | 0.635 | 1814 | 0.274 | 5942 | 0.591 | 2514 | 0.604 |
> | SFT | 0.750 | 1515 | 0.265 | 6547 | 0.601 | 2479 | 0.553 |
> | RL | 0.808 | 639 | 0.401 | 1881 | 0.657 | 863 | 0.581 |
> | LUFFY | 0.804 | 1168 | **0.441** | 3285 | 0.652 | 1516 | 0.572 |
> | SFT then RL(v1) | 0.723 | 879 | 0.289 | 3779 | 0.562 | 1217 | 0.548 |
> | SFT then RL(v2) | 0.728 | 980 | 0.339 | 4490 | 0.620 | 1501 | 0.627 |
> | ReLIFT | **0.816** | 1321 | 0.431 | 3177 | **0.670** | 1511 | **0.643** |
>
> ReLIFT performs strongly across these benchmarks, achieving the highest average score and ranking either first or second on every single one. This underscores the robust generalization ability enabled by ReLIFT's integration of RL and fine-tuning, regardless of whether the tasks involve code, STEM, or other domains.
>
> Notably, pure RL demonstrates great performance on both ARC-Challenge and MMLU-Redux, thanks to its superior OOD generalization—other method combining RL and SFT perform noticeably worse. However, ReLIFT, although incorporating external supervision, still surpasses pure RL on these specific benchmarks.

---

> ### Author Response · Authors · 2025-11-20
> **Author Response**
>
> ### Response to W3
>
> >There’s no ablation on using thresholds like acc(q) ≤ p, or on how many “hardness levels” might be useful.
>
> We sincerely thank your valuable feedback about the limited ablation on different thresholds. As reviewer u4oo and tK9a all suggest ablation experiments, we conduct a detailed ablation study on two key hyperparameters of ReLIFT: the fine-tuning buffer threshold (M) and the difficulty threshold (Q). The difficulty threshold $Q$ acts as the gating function, selectively determining which degree of incorrect samples (accuracy<=$Q$) are retained for SFT. $M$ controls the maximum size of this fine-tuning buffer. We evaluate the average performance of different M and Q combinations on the five math benchmarks within 400 training steps. The experimental results are shown in the table below. The full results are listed in Appendix H.
>
> | Q    | M=32  | M=64  | M=128 | M=256 |
> |------|-------|-------|-------|-------|
> | 0    | 0.411 | 0.490 | 0.478 | 0.459 |
> | 1/8  | 0.405 | 0.404 | 0.493 | 0.471 |
> | 1/4  | 0.392 | 0.403 | 0.467 | 0.491 |
>
> * Analysis of Gating Hyperparameters $Q$ and $M$: We observe that extreme settings for the key hyperparameters—the difficulty threshold $Q$ and the buffer size $M$—lead to suboptimal results. Specifically, when the buffer size $M$ is overly small (e.g., $M=32$) or when a relaxed filtering condition $Q$ is combined with a small $M$ (e.g., $Q=1/4$ with $M=64$), the model's performance significantly degrades (down to $0.392$ and $0.403$). This clearly indicates that either an small size buffer or an overly relaxed filtering condition results in too frequent SFT updates, failing to provide the necessary stability during RL.
>
> * Robust Performance Configurations and Recommended Setting: The ablation results reveal the existence of several robust parameter configurations that yield peak performance (approximately $0.49\pm0.003$):$(M=64, Q=0)$, $(M=128, Q=1/8)$ and $(M=256, Q=1/4)$. This demonstrates the robustness of ReLIFT to the parameters $Q$ and $M$, with multiple suitable hyperparameter choices available.
>
> * The configuration $(M=64, Q=0)$ is still our experiment setting because the $Q=0$ condition demands the minimum amount of demonstration data. Furthermore, as shown in the ablation study (Figure 4), utilizing SFT samples from less difficult questions leads to a notable increase in token length by overriding the model's concise response patterns for simpler problems. Thus, $(M=64, Q=0)$ successfully achieves near-optimal performance while minimizing the complexity of the gating function and mitigating undesirable token length expansion.
>
> ### Response to Q1
>
> >A reliable evaluation on Math and OlympiadBench
>
> We appreciate your concern regarding the stability of evaluation results, especially given the inherent randomness during inference. To address this important issue, we revise our evaluation methodology for the MATH and OlympiadBench. Originally, following the LUFFY protocol, we reported only the mean performance over a single response (mean@1). We now adopt an improved approach by computing the mean across eight responses (mean@8). Notably, the shift from mean@1 to mean@8 results in only a minor change in the reported statistics—approximately 1%—and the strong performance of ReLIFT remains consistently evident across both benchmarks. The updated final results for MATH and Olympiad have been incorporated into the PDF accordingly.
>
> We hope our responses answer your questions, and again, thanks for your advice to improve our paper!
>
> [1] Clark, P. et al. Think you have Solved Question Answering? Try ARC, the AI2 Reasoning Challenge. *arXiv:1803.05457v1*, 2018.
>
> [2] Rein, D. et al. Gpqa: A graduate-level google-proof q\&a benchmark. *First Conference on Language Modeling*, 2024.
>
> [3] Gema, A.P. et al. Are We Done with MMLU?. *arXiv:2406.04127*, 2024.
>
> [4] Chen, M. et al. Evaluating Large Language Models Trained on Code. *arXiv:2107.03374*, 2021.

---

### Meta-Review · Area_Chair_w4Ly · 2026-01-05

**Summary:**

This paper introduces ReLIFT, a training strategy that interleaves reinforcement learning with targeted supervised fine-tuning on questions the model fails to solve during rollouts. All four reviewers acknowledge the paper's clear presentation, well-motivated analysis of RL versus SFT training dynamics, and the simplicity of the proposed method. The primary concerns raised during review, such as limited out-of-distribution evaluation, missing ablation studies on key hyperparameters, and questions about response length adaptivity, were substantively addressed in the rebuttal through additional experiments on new benchmarks, comprehensive ablations, and detailed length-accuracy analyses. Reviewer tK9a raised their score from 4 to 6 following the rebuttal, and Reviewer H2Ni explicitly acknowledged the responses as strengthening the paper's claims. Some scope limitations remain: the method fundamentally relies on access to high-quality demonstrations from stronger models, and the "online" framing is somewhat undermined by the offline data collection in experiments. However, the consistent empirical improvements across diverse benchmarks, the resource efficiency gains, and the practical utility of the approach outweigh these concerns. I recommend acceptance.

**Reviewer Concerns:**

Addressed Concerns

OOD evaluation (z9hW, u4oo, tK9a): Authors added new benchmarks covering code, STEM, and general reasoning

Ablation studies (z9hW, u4oo, tK9a): Comprehensive experiments on buffer size and difficulty threshold were provided

Response length concerns (u4oo): Length-accuracy analysis showed ReLIFT adapts appropriately rather than over-penalizing long reasoning

Training stability (tK9a): Added training curves comparing ReLIFT to LUFFY showing more stable optimization

Evaluation reliability (z9hW): Switched from mean@1 to mean@8 for more robust evaluation

Missing baselines (H2Ni): Added SFT with entropy regularization and expanded Llama baselines

Outstanding Concerns

Scope/applicability: The method still fundamentally depends on access to high-quality demonstrations from stronger models; the claim of "pushing boundaries" versus "efficient distillation" remains somewhat unresolved

Offline vs. online data: Authors acknowledged the mismatch between the "online" motivation and their offline experimental setup, offering only a theoretical argument that true online collection is feasible

Generalization to strongest models: Whether ReLIFT benefits GPT-4-level models remains hypothetical with no empirical evidence

**Reviewer Scores:**

Reviewer z9hW (6): Likely would have remained at 6. Their empirical concerns (OOD evaluation, ablations, evaluation methodology) were thoroughly addressed, but their fundamental concern about scope: whether this is truly "pushing boundaries" versus efficient distillation, was only partially addressed with discussion rather than new evidence.

Reviewer u4oo (6): Likely would have increased the score. All three of their specific concerns (OOD benchmarks, gating function ablations, length-accuracy analysis) received direct experimental responses. They did not engage with the rebuttal, but the authors' responses were comprehensive and on-point.

Reviewer H2Ni (8): Would have remained at 8. Already the strongest advocate; explicitly acknowledged the responses as satisfactory and stated the additions "will strengthen the paper's claims." No outstanding concerns.

Reviewer tK9a (4→6): Already raised their score after the rebuttal and thanked the authors. Their core concerns about ablations and limited improvements were addressed.

---

### Decision · Program_Chairs · 2026-01-26

Accept (Poster)